# Modular Federated Contrastive Learning with Twin Normalization for Resource-limited Clients

**Azadeh Motamedi**  *19am43@queensu.ca*
*Department of Electrical and Computer Engineering*
*Queen's University*

**Il-Min Kim**  *ilmin.kim@queensu.ca*
*Department of Electrical and Computer Engineering*
*Queen's University*

**Reviewed on OpenReview:** *https://openreview.net/forum?id=FlxnywWi14*

## Abstract

Despite recent progress in federated learning (FL), the challenge of training a global model across clients, having heterogeneous, class-imbalanced, and unlabeled data, is not fully resolved. Self-supervised learning requires deep and wide networks, and federal training of those networks induces a huge communication/computation burden on the client side. We propose Modular Federated Contrastive Learning (MFCL) by changing the training framework from end-to-end to modular, meaning that instead of federally training the entire network, only the first layers are trained federally through a server, and other layers are trained at another server without any forward/backward passes between servers. We also propose Twin Normalization (TN) to tackle data heterogeneity. Results show that ResNet-18 trained with MFCL(TN) on CIFAR-10 achieves 84.1% accuracy when data is severely heterogeneous while reducing the communication burden and memory footprint compared to end-to-end training. The code will be released upon paper acceptance.

## 1 Introduction

Federated Learning (FL) has been introduced as a framework to learn from the data located at multiple clients without transferring the client's raw data to a central location, called the server McMahan et al. (2016; 2017); Konečnỳ et al. (2016a;b). In each FL round, each client's local model is trained on its dataset, and the local models' parameters are sent to the server to be aggregated and broadcast back to the clients. The distribution of local datasets plays a vital role in orchestrating the real-world implementations of FL. Simply aggregating the model parameters trained on heterogeneous datasets does not improve the global model accuracy unless additional tricks are adopted such as regularization techniques in local objectives of the clients Luo et al. (2021); Dieuleveut et al. (2021); Mu et al. (2021); Li et al. (2020; 2021); Durmus et al. (2021); Yu et al. (2020b), adaptive aggregation in the server Wang et al. (2020a), data sharing on the clients or server Zhao et al. (2018); Hao et al. (2021), and data augmentation Shin et al. (2020) to reduce the gap between the local and global models. Meanwhile, addressing the challenges of data labelling in FL, some self-supervised methods have been proposed considering data heterogeneous clients Lubana et al. (2022); Lu et al. (2022); Makhija et al. (2022); Huang et al. (2022); Miao & Koyuncu (2022); Yu et al. (2020a). In particular, contrastive Chen et al. (2020) and non-contrastive learning Grill et al. (2020); Chen & He (2021) methods have been applied to FL Zhang et al. (2020b); Zhuang et al. (2021; 2022). The existing techniques are mostly based on training one or even two deep networks at each client, adopting various methods such as moving average update at local models Zhuang et al. (2022), tuning the communication frequency between the clients and server based on the divergence of local and global models Zhuang et al. (2021), or aggregating and sharing the clients' representations based on their distance Zhang et al. (2020b) to ensure that clients can *collaboratively* learn from the unlabeled data Shi et al. (2021). For a detailed review, see Appendix A.1.

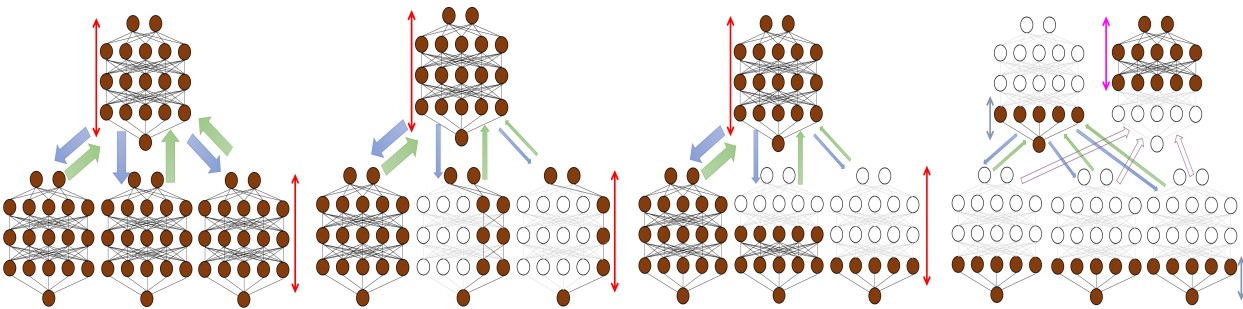

(a) E2E traditional FL (supervised/self-supervised) Konečnỳ et al. (2016a;b).

(b) E2E resource compatible FL (supervised) Horvath et al. (2021).

(c) E2E resource compatible FL (supervised) Kim et al. (2022); Hong et al. (2022).

(d) Modular resource compatible FL (self-supervised) [ours].

Figure 1: Existing resource-compatible schemes and the proposed method. The red arrows indicate the forward and backward passes through all layers. The parameters of the bold connections are transmitted to the server in each FL round, as indicated by the blue and green arrows. (a) Clients send *all* parameters to the server in *end-to-end* (*E2E*) FedAvg. (b) Clients send parameters of *some filters* (a vertical cut) of the model to the server in *E2E* HeteroFL. (c) Clients send parameters of *some layers* (a horizontal cut) of the model to the server in *E2E* Split-Mix. (d) Clients send parameters of *some layers* (a horizontal cut) of the model to the first server (gServer) in the proposed *modular* scheme. After the initial layers are trained, the rest of the model is trained on the representations received in the rServer. The gray and pink arrows show the forward and backward passes on the client side and rServer, respectively. The white arrows show the transmission of the representation vectors, which is done only once.

Typically, self-supervised learning relies on a large amount of data, deep and wide models, and/or large batch sizes to achieve good performance, which means high computing power and/or large memory footprints are needed Zhuang et al. (2022). Such resources are often unavailable on clients' devices, especially in cross-device FL scenarios. To address these issues, researchers have proposed different *scaling methods* for clients to have different-sized local models depending on their available computation resources (Figure 1). In Diao et al. (2020); Horvath et al. (2021), the authors suggested dividing the network based on its *width* and assigning a different number of *filters* to different clients based on their computational resources (Figure 1b). In more recent schemes, the network is divided based on its *depth*, and different numbers of modules (sub-models) are assigned to different clients Kim et al. (2022); Hong et al. (2022) (Figure 1c). However, there are mainly three limitations in these resource-compatible architectures. First, in those schemes, only supervised loss functions are considered. *Shallow* or *narrow* networks are typically insufficient for self-supervised tasks because they lack the capacity to learn rich, informative representations from complex or highly diverse data, particularly in scenarios involving heterogeneous or class-imbalanced data. In FL, where data heterogeneity is a major challenge, shallow networks are unlikely to extract representations that generalize well across different clients, especially when each client has unique data distributions. Self-supervised learning methods, either contrastive Chen et al. (2020) or non-contrastive Grill et al. (2020), require networks with enough depth and capacity to align representations from diverse data sources effectively.

Second, the performance of those schemes has not been comprehensively studied under severe heterogeneous and class-imbalanced (CIB) data scenarios. If some clients with unique data cannot afford to train a deep and wide sub-model, the global model cannot learn those unseen distributions. The performance degradation might be even worsened when the local data is CIB or the number of clients is large (e.g., more than 10) Hong et al. (2022). Third, although those schemes are proposed for resource-limited clients, the computation (and communication) burden on the side of clients is still high Kim et al. (2022). Moreover, clients still need large *memory* to implement the entire graph to backpropagate the gradients from the last layer to the first layer of the whole network, which is needed for any type of end-to-end (E2E) learning.

Although E2E learning performs well on numerous tasks, in some cases (e.g., ill-conditioned problems), E2E learning might converge slowly, converge into possibly local optima, face vanishing gradients Shalev-Shwartz

et al. (2017), or be expensive/difficult to implement due to its serial nature. As an alternative approach to E2E learning, modular learning has recently been studied Pfeiffer et al. (2023); Belilovsky et al. (2019); Wang et al. (2021b), in which a model is split into multiple modules, which are trained on distinct loss functions and the gradients of each module are not necessarily backpropagated to the other modules. Existing modular learning methods can be categorized into two groups from the perspective of gradient backpropagation (BP) (i) Brain-inspired BP-free approach Illing et al. (2021) that trains the network based on Hebbian rules and (ii) BP-limited approach Löwe et al. (2019) that limits BP to a few modules using various techniques.

In this paper, inspired by modular learning, and considering the restrictions of communication, computation and memory resources on the side of clients for adopting self-supervised learning, we propose a new and effective FL method, namely Modular Federated Contrastive Learning (MFCL). In MFCL, instead of federally training an entire model across the clients, the model is split (at a cut-off layer) into two modules that are separately trained (Figure 1d). The layers up to the cut-off layer are called the *client module*. This module is federally trained across clients through the gradients server or the *gServer*. The remaining layers are called the representations server module or the *rServer module*. The rServer module is trained at the *rServer*.[1] The data representations are extracted from the clients' unlabeled data by the federally trained clients' module and transmitted to the rServer. The rServer module is trained based on self-supervised contrastive learning on the representations received from the clients at the rServer. In MFCL, the two modules are trained separately (through two distinct servers) without any forward/backward passes between them, which is essentially the second type of modular learning (i.e., the BP-limited approach).

To address the issue of data heterogeneity in FL, we propose a new normalization method, namely Twin Normalization (TN). Batch Normalization (BN) fails when the mini-batch data is CIB [Wu & He (2018)]. The performance of BN can further deteriorate in FL under severe heterogeneous data Diao et al. (2020). TN, unlike BN, is a *sample-based normalization* which normalizes *two augmented views* or *two examples* of *one sample* and doesn't rely on mini-batches size, which makes it more efficient in heterogeneous CIB data. Specifically, TN is designed to avoid some of the key drawbacks of BN in the context of FL, especially in heterogeneous and CIB scenarios. Similar to Group Normalization (GN) Wu & He (2018), TN normalizes across a group of channels, making it robust in scenarios with small mini-batches. However, compared to GN, TN leverages more diverse information within each group of channels, as it normalizes the channels of two augmented views of the same sample together. This results in more accurate mean statistics for those groups, enhancing the robustness of the normalization. TN effectively captures both inter-example and intra-example relationships, improving model performance. Our contributions are as follows:

- Considering the resource constraints of clients, we propose a modular framework to perform contrastive learning on the server. MFCL reduces communication overhead and allows the rServer to conduct contrastive learning on more class-balanced (CB) data, improving the model's generalization.

- The proposed MFCL addresses the issue of limited memory and computing resources at the client devices by shifting the resource-intensive contrastive learning to the Server. This makes the system more feasible for real-world deployment on resource-constrained clients.

- We propose TN, for contrastive learning in FL, which performs better than BN and Group Normalization (GN) in severe data heterogeneous scenarios.

- Through experiments, we demonstrate the effectiveness of the proposed MFCL, especially with TN, which achieves robust, stable, and state-of-the-art performance on severe heterogeneous and CIB data while only a small-size client module is trained federally across clients.

---

[1]In FL literature, leveraging two (or more) servers has been demonstrated to be useful for addressing (i) data heterogeneity Caldarola et al. (2021); Ghosh et al. (2020), (ii) data privacy Thapa et al. (2022); Li et al. (2022), (iii) communication and computation efficiency Khan et al. (2021); Liu et al. (2020a); Wang et al. (2019), and (iv) personalization Mansour et al. (2020). In MFCL, adopting two servers ensures data privacy, which will be later discussed.

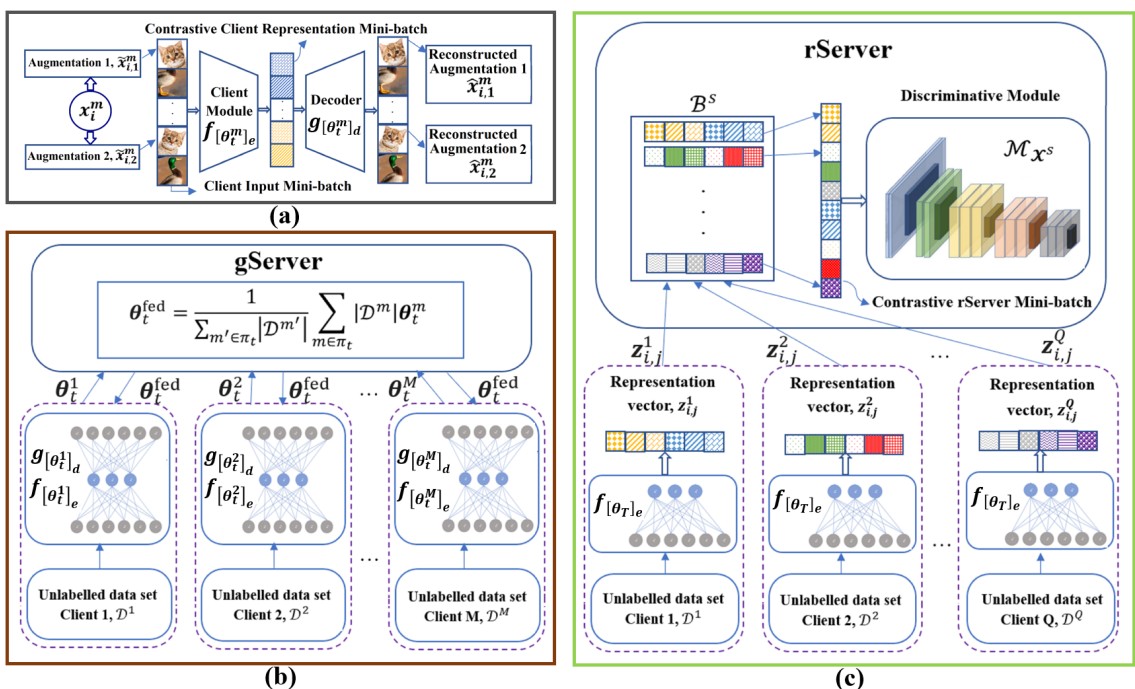

Figure 2: (a) Client module and decoder. Client $m$ creates two augmented versions $\tilde{\mathbf{x}}_{i,j}^m$, $j = 1, 2$ of $\mathbf{x}_i^m \in D^m$ and constructs contrastive client input mini-batches $B^m = \left(\tilde{\mathbf{x}}_{1,1}^m, \ldots, \tilde{\mathbf{x}}_{K,1}^m, \tilde{\mathbf{x}}_{1,2}^m, \ldots, \tilde{\mathbf{x}}_{K,2}^m\right)$. (b) Training phase 1: Client modules/decoders are trained federally across clients through the gServer. (c) Training phase 2: Clients generate representations $\mathbf{z}_{i,j}^m = f_{[\boldsymbol{\theta}_T]_e}(\tilde{\mathbf{x}}_{i,j}^m)$, $j = 1, 2$ from the trained client module and send them to rServer. The rServer constructs $B^s$ from the received representations and updates its parameters $\chi$ by minimizing contrastive loss.

## 2 Proposed Modular Federated Contrastive Learning

In the literature, there are a few works on contrastive learning-based FL, and in those works, the contrastive loss is optimized at each client device. However, considering the memory and computing resource limitations of the clients, it might be difficult or, at times, even impossible to perform contrastive learning at the clients Li et al. (2014), because the contrastive loss typically benefits from the larger model, larger batch size, and/or longer training time Chen et al. (2020); Zhuang et al. (2022). Besides, directly optimizing the contrastive loss across clients as in the traditional FL (i.e., averaging all parameters of clients' local models trained with contrastive loss) may filter out the subtle yet useful information of the difference between the weights corresponding to the augmented versions of data, which is important in determining the similarities of examples. To address these issues, in our proposed MFCL, the contrastive loss is optimized at the rServer. Specifically, instead of federally training the entire deep network across clients on a contrastive loss, only a part of the model is trained by a contrastive loss at the rServer on data representations received from the clients' unlabeled data extracted by the client module. The client module at the gServer and the rServer module at the rServer are trained separately without any forward/backward passes between them.[2]

### 2.1 Training Procedure for the Proposed MFCL

*Training Phase 1 for Client Modules:* During the first stage of training, the client modules are federally trained over $M$ clients with coordination of the gServer. The client module, denoted by $f_{[\boldsymbol{\theta}_t^m]_e}(\cdot)$, consists of the first several layers up to a cut-off layer of the entire model. Here, $[\boldsymbol{\theta}_t^m]_e$ denotes the parameters of the client module at client $m$ at FL round $t$. For unsupervised (federated) training of the client module, we

---

[2]In this work, we assume the two servers do not communicate (or, do not collude).

introduce the decoder $g_{[\boldsymbol{\theta}_t^m]_d}(\cdot)$ as a mirror structure of the client module, with a proper loss (e.g., cross-entropy or mean squared error (MSE)), as shown in Figure 2(a). Here, $[\boldsymbol{\theta}_t^m]_d$ denotes the parameters of the decoder at client $m$ at FL round $t$.[3] For normalizing the layers' outputs of the client module and its decoder, one could use the widely adopted BN and it *would* work *if* the clients' data *were* homogeneous and CB. However, in practical FL, the clients' data are heterogeneous/CIB, and thus, BN often does not work well. To address this issue, given the special structure of the contrastive mini-batch at the clients, we propose using a TN layer after each convolutional layer in the client module and after each transposed convolution layer in the decoder. The details about TN are provided in the next section. The architectures of the client module and its decoder for ResNet-18 and ResNet-50 are described respectively, in Tables 9 and 10 in the Appendix.

At the beginning of the first FL round, the gServer sends the initial parameters to a set of selected clients. These clients train their modules on their datasets. Specifically, at the end of FL round $t-1$, the selected clients send the updated parameters, $\boldsymbol{\theta}_{t-1}^m, m \in \pi_{t-1}$, to the gServer, and the gServer aggregates the parameters as $\boldsymbol{\theta}_{t-1}^{\text{fed}} = \frac{1}{\sum_{m' \in \pi_{t-1}} |D^{m'}|} \sum_{m \in \pi_{t-1}} |D^m| \boldsymbol{\theta}_{t-1}^m$.[4] In FL round $t$, the gServer broadcasts the aggregated parameters, $\boldsymbol{\theta}_{t-1}^{\text{fed}}$, to a set $\pi_t$ of clients. Each client $m \in \pi_t$ locally updates the received averaged parameters $\boldsymbol{\theta}_{t-1}^{\text{fed}}$ to $\boldsymbol{\theta}_t^m$ with the following objective:

$$\min_{\boldsymbol{\theta}_t^m} \mathbb{E}_{\mathbf{x} \sim D^m} \left[ \mathscr{L}_{\boldsymbol{\theta}_{t-1}^{\text{fed}}} \left( \mathcal{A}(\mathbf{x}), g\left(f\left(\mathcal{A}(\mathbf{x})\right)\right) \right) \right], \tag{1}$$

where $\mathscr{L}_{\boldsymbol{\theta}_{t-1}^{\text{fed}}}(\cdot, \cdot)$ is the loss function to optimize $\boldsymbol{\theta}_t^m$ and $\mathcal{A}(\cdot)$ denotes data augmentation as follows. For training the client modules/decoders, client $m$ first creates two augmented versions $\tilde{\mathbf{x}}_{i,j}^m, j = 1, 2$ of $\mathbf{x}_i^m \in D^m$, where $\mathbf{x}_i^m$ denotes the $i$-th data sample of the local dataset $D^m$ of client $m$. Combining the augmented versions of $K$ data samples, client $m$ constructs *the contrastive client input mini-batches* of size $2K$ denoted by $B^m = \left( \tilde{\mathbf{x}}_{1,1}^m, \ldots, \tilde{\mathbf{x}}_{K,1}^m, \tilde{\mathbf{x}}_{1,2}^m, \ldots, \tilde{\mathbf{x}}_{K,2}^m \right)$.

*Training Phase 2 for the rServer Module:* The second stage begins when federated training of client modules and their decoders is finished, say at FL round $T$. Then, the decoder is discarded and, supplying two augmented versions $\tilde{\mathbf{x}}_{i,j}^m, j = 1, 2$, of each data sample $\mathbf{x}_i^m \in D^m$ to the trained client module, client $m$ produces representation vectors, $\boldsymbol{z}_{i,j}^m = f_{[\boldsymbol{\theta}_T]_e}(\tilde{\mathbf{x}}_{i,j}^m), j = 1, 2$, called the *contrastive client representation mini-batches*, and transmits them to the rServer. Combining the representations received from at least $Q \leq M$ clients, the rServer constructs the contrastive rServer mini-batches, $B^s$, such that each mini-batch $B^s$ is large enough for contrastive learning. In general, the rServer representations are much more CB, than the data at each client (see Figure 6 in the Appendix). At the rServer, the rServer module, $\mathscr{M}_{\boldsymbol{\chi}^s}$, parameterized by $\boldsymbol{\chi}^s$, is trained by minimizing the contrastive loss on contrastive rServer mini-batches, $B^s$ (see Appendix A.2). Figures 2(b) and 2(c) show the two training phases of MFCL, respectively.

Note that MFCL is designed to result in a single global model. To that end, when the training is done, the client module and the rServer module form a single unified global model. To deploy this unified model, the (finally trained) rServer module should be broadcast to the clients. Algorithm 1 outlines the detailed process of the MFCL framework.

## 3 Normalization for the Client Module/Decoder

Without any normalization, CIB data in the mini-batch can increase the variance of the mini-batch gradients. Therefore, the output distribution of layers may diverge, which leads to lower accuracy Wu & He (2018). In the following, we first briefly discuss BN and GN. We then propose our normalization method, TN.

---

[3]For simplicity, the model parameters of the whole autoencoder (i.e., the client module and it's decoder) are denoted by $\boldsymbol{\theta}_t^m$. $[\boldsymbol{\theta}_t^m]_e$ and $[\boldsymbol{\theta}_t^m]_d$ represent the model parameters of the client module and decoder parts of $\boldsymbol{\theta}_t^m$, respectively.

[4]We used FedAvg McMahan et al. (2016) to aggregate the clients' module parameters at the gServer. Other hyperparameters and augmentation techniques are detailed in Tables 12 and 13 in the Appendix.

### 3.1 Batch Normalization (BN)

The strong point of BN is that each channel is normalized over the entire mini-batch Ioffe & Szegedy (2015). BN works well in contrastive learning when each contrastive mini-batch is large and CB (this is the case for the contrastive mini-batch at the rServer). However, when the mini-batch size is smaller and/or the data of each mini-batch is CIB, the batch statistics estimated by BN are biased and inaccurate. Therefore, using BN for training the client modules/decoders on the heterogeneous and CIB data in FL leads to biased models, which are unable to accurately learn the features of those classes having only a small number of samples.

### 3.2 Group Normalization (GN)

The dependence of the batch-normalized outputs of each layer on the entire mini-batch makes BN powerful; however, it is also the source of BN drawbacks. GN, proposed for distributed training, normalizes a group of channels of a sample together without utilizing the entire mini-batch Wu & He (2018). Specifically, GN *randomly* chooses a group of $\Lambda \le C$ channels, which may have different scales, and normalizes them together. GN is applied to FL to address the issue of heterogeneous data across clients and works better than BN Zhang et al. (2021); Yu et al. (2021; 2020b). The reason is that in FL, the model parameters (e.g., the filters' parameters in convolutional neural networks) are averaged over multiple clients. Averaging makes the scales of filters smoother. Applying smoother filters on channels leads to smoother features and normalizing a sample over a group of its channels, as in GN, becomes meaningful. One limitation of GN is that it relies on limited information to normalize each sample over a group of $\Lambda$ channels, compromising the precision in estimating the statistics of each sample to make the statistics independent of the mini-batch size.

### 3.3 Twin Normalization (TN)

We propose TN as a solution to address the lack of sufficient information in GN for accurately estimating the statistics of a sample across a group of its features. Since in contrastive learning, there are two augmented views of each sample, it is possible to benefit from the rich and diverse information of those two examples and normalize them together. TN normalizes positive examples together, which increases their similarity and further differentiates them from negative ones. For mathematical formulation, we let $\boldsymbol{w}_{i,j}^{c,l}, i = 1, \ldots, K; j = 1, 2; c = 1, \ldots, C_l$ denote the attributes of channel $c$ at layer $l$ for the $j$-th augmented version, $\tilde{\mathbf{x}}_{i,j}^m, j = 1, 2$, of the $i$-th input data sample, $\mathbf{x}_i^m$, in the client's mini-batch. We first determine the mean $\boldsymbol{\mu}_i^{\Phi_l,l}$ and variance $(\boldsymbol{\sigma}_i^{\Phi_l,l})^2$ of $\Lambda$ channels corresponding to augmented versions of each input in the $l$-th layer as follows:

$$\boldsymbol{\mu}_i^{\Phi_l,l} = \frac{1}{2\Lambda} \sum_{j=1}^{2} \sum_{c \in \Phi_l} \boldsymbol{w}_{i,j}^{c,l} \quad \text{and} \quad (\boldsymbol{\sigma}_i^{\Phi_l,l})^2 = \frac{1}{2\Lambda} \sum_{j=1}^{2} \sum_{c \in \Phi_l} \| \boldsymbol{w}_{i,j}^{c,l} - \boldsymbol{\mu}_i^{\Phi_l,l} \|^2 \tag{2}$$

where $\| \cdot \|$ denotes the $l_2$ norm and $\Phi_l \subseteq \{1, \ldots, C_l\}$ denotes the set composed of $\Lambda$ channels at layer $l$. Then, we perform the normalization as follows:

$$\bar{\boldsymbol{w}}_{i,j}^{c,l} = \bar{\boldsymbol{\theta}}_t^{m,l} \frac{\boldsymbol{w}_{i,j}^{c,l} - \boldsymbol{\mu}_i^{\Phi_l,l}}{\sqrt{(\boldsymbol{\sigma}_i^{\Phi_l,l})^2}} + \tilde{\boldsymbol{\theta}}_t^{m,l}, \quad \text{for } c \in \Phi_l, \tag{3}$$

where $\bar{\boldsymbol{w}}_{i,j}^{c,l}$ is normalized $\boldsymbol{w}_{i,j}^{c,l}$ for $c \in \Phi_l$. Also, $\bar{\boldsymbol{\theta}}_t^{m,l}$ and $\tilde{\boldsymbol{\theta}}_t^{m,l}$ denote trainable scaling factor and shift, respectively, for layer $l$ of the client's module and decoder at FL round $t$.

TN ensures that the attributes of examples are normalized independently of other examples in the client's contrastive mini-batch. The pros of adopting TN are as follows: (1) Unlike BN, TN is not sensitive to mini-batch size, making it more suitable for FL where mini-batches can be small and vary across clients. (2) TN's ability to normalize augmented views together enhances the similarity between positive examples and separates them from negative examples, which is crucial in contrastive learning. This improves the generalization of models trained in FL with heterogeneous CIB data.

Although TN works well with heterogeneous CIB data, BN still works better than TN when the data is homogeneous and balanced. Therefore, for training the rServer module at the rServer, we can still benefit

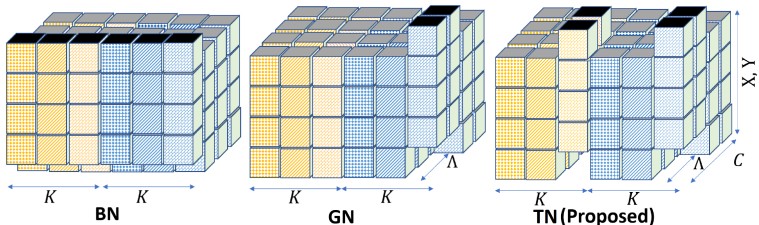

Figure 3: Normalization techniques for the client module/decoder of each client. The $x$-axis, denoted by $2K$, represents the elements of each contrastive client mini-batch. The $y$-axis, denoted by $C$, represents the channels and the $z$-axis, denoted by $(X, Y)$, represents the spatial axes of images' pixels. Only the misaligned blocks with black tops are simultaneously normalized together with the same mean and variance.

from BN Ioffe & Szegedy (2015). This is because, compared to each client's contrastive input/representation mini-batches, the rServer's contrastive representation mini-batches can be constructed as large as needed and much more resources are typically available at the rServer. Thus, the contrastive rServer mini-batches are much better CB and large enough to warrant BN.

### 3.4 The comparison between BN, GN, and TN

With some inspiration from BN and GN, TN is uniquely adapted for federated contrastive learning. Specifically, BN relies on the whole mini-batch, and the size of the mini-batch plays an important role in the success of BN. TN is designed to avoid some of the key drawbacks of BN in the context of FL, especially in heterogeneous and CIB scenarios. TN, unlike BN, is a sample-based normalization which normalizes two examples of *one sample* and doesn't rely on mini-batches size, which makes it more efficient in class-imbalanced and heterogeneous data settings. GN normalizes across a group of channels, making it robust in scenarios with small mini-batches. Compared to GN, TN has more diverse information per group of channels as TN normalizes the group of channels of two augmented views of the same sample together, making the mean of those groups more accurate. TN effectively captures both *inter-example* and *intra-example* relationships, improving model performance. In neither BN nor GN, two examples of one sample are normalized together as in TN. Figure 3 shows the differences among BN, GN, and TN.

In summary, while BN struggles with heterogeneous data due to its reliance on batch-wide statistics, and GN, though more robust, lacks inter-example normalization, TN combines the strengths of both. TN retains GN's independence from mini-batch size while having inter-example (not inter-sample) relationships, in a controlled manner by focusing on augmented pairs. This makes TN particularly well-suited for federated contrastive learning in heterogeneous environments, where it improves model performance without suffering from the drawbacks of BN and GN.[5]

## 4 MFCL vs Federated Split Learning

### 4.1 Methodological Comparison

The proposed MFCL might appear to be similar to Federated Split Learning (FSL) Thapa et al. (2022) in that we also split the network at a cut-off layer into the client module and the rServer module. However, MFCL and FSL differ significantly in major aspects. First, in FSL, the smashed data (i.e., the output of the cut-off layer) must be communicated between the clients and server as many times as the number of FL rounds. In contrast, in MFCL, the data representations are transmitted from the clients to the rServer only a single time, after the clients' module and decoder are trained. Therefore, MFCL imposes a significantly lower data traffic burden compared to FSL. Second, unlike FSL which backpropagates the gradients of the smashed data from the server to the clients during the training, MFCL trains two separated modules without

---

[5]Note that while TN is particularly beneficial in federated contrastive learning, it is not necessarily limited to this context. TN could be extended to any learning tasks that benefit from augmented data samples and the data is CIB.

Table 1: The accuracy performance of MFCL with various cut-off layers for ResNet-18 on CIFAR-10 with $M = 10$ clients, and ResNet-50 on Tiny-ImageNet with $M = 100$ clients ($\xi = 0.1$).

|  | 1 | 2 | 3 | 4 | 5 | 6 | 7 |
|---|---|---|---|---|---|---|---|
| ResNet-18 (on CIFAR-10) | 59.1 | **84.5** | 82.4 | 74.3 | 61.5 | 52.3 | 47.6 |
| ResNet-50 (on Tiny-ImageNet) | 25.8 | 30.6 | 32.7 | **33.9** | 33.7 | 32.2 | 28.5 |

any gradient/weight transactions. Last, but not least, MFCL can be more advantageous than FSL in terms of data privacy, which is discussed in what follows.

### 4.2 Data Privacy Comparison

In MFCL, *only the representations* of the harshly augmented data are transmitted (disclosed) *only* to the rServer. This means that the gServer has access *only* to the parameters of the client modules/decoders, not the representations produced by the client modules, whereas the rServer has access *only* to the data representations, not the parameters of the clients' modules/decoders. In FSL, the server has access only to a part of the model and this setup provides a certain level of data privacy Wang et al. (2023b); Turina et al. (2020); Jeon & Kim (2020); Thapa et al. (2021); Li et al. (2022). Comparing MFCL with FSL, MFCL offers more enhanced data privacy than FSL due to its distinct training phases on two distinct servers. On top of that, in MFCL, the data representations are sent to the rServer only a single time. In contrast, FSL entails the transmission of smashed data in every FL round throughout the training process, which makes it more vulnerable to data leakage. Moreover, in MFCL, the client module/decoder and gServer do not have any gradient transactions with the rServer, which has access to the data representations, ensuring that the rServer does not have the gradients of representations. By contrast, in FSL, the server has smashed data as well as the corresponding gradients, making the whole system even more vulnerable. MFCL bolsters data privacy by reducing the risks associated with data leakage.

## 5 Experiments

### 5.1 Setup and Performance Metric

We perform our experiments on CIFAR-10, CIFAR-100 Krizhevsky (2009), and Tiny-ImageNet Le & Yang (2015) with ResNet-18 and ResNet-50. For ResNet-18, the first two layers, and for ResNet-50, the first 4 layers are chosen as the client module (i.e., the encoder part) and we design the decoder part as a mirror structure of the encoder. Table 1 presents the results of choosing the cut-off layer at different locations for the two networks. In both cases, the client module and its decoder are composed of convolutional layers and TN layers (See Tables 9 and 10 in the Appendix for more details). The rServer module is composed of the remaining layers of the network, which is trained on the received representations with contrastive loss.

We implemented MFCL with TensorFlow 2.14 following the standard structure of FL McMahan et al. (2017) and the contrastive learning Chen et al. (2020). To construct heterogeneous and CIB datasets over the clients, we used Dirichlet distribution, with concentration parameter ($\xi > 0$) Yurochkin et al. (2019); Wang et al. (2020b). If $\xi$ is set to a smaller value, the data becomes more heterogeneous and CIB, whereas when $\xi \to \infty$, the data becomes homogeneous and CB (See Figure 7 in the Appendix). We used a single NVIDIA GeForce RTX 3090 GPU to simulate the clients and rServer modules. We evaluated the classification accuracy of MFCL based on the linear evaluation method Chen et al. (2020). Specifically, we used non-linear projection layers as described in Table 11 in the Appendix on top of the rServer module, and one linear Dense layer on top of the projection layers for linear evaluation. In training phase 2, the rServer module, projection layers, and linear (classifier) layer can be jointly trained as long as the gradients are not backpropagated from the linear (classifier) layers to the rServer module and the projection layers. Alternatively, the rServer module and the projection layers could be trained first, and then, the Dense layer on top of the frozen (trained) rServer module and projection layers can be fine-tuned using labelled data.

Table 2: Accuracy comparison of MFCL and baselines on CIFAR-10 with $M = 10$ on ResNet-18, CIFAR-100 with $M = 100$ on ResNet-18, and Tiny-ImageNet with $M = 100$ on ResNet-50. The best performance is highlighted in boldface. For SSFL methods, the top-1 accuracy with linear evaluation is reported. For MFCL, BN is used at the rServer.

| Method \ $\xi$ | CIFAR-10 | | | CIFAR-100 | | | Tiny-ImageNet | | |
|---|---|---|---|---|---|---|---|---|---|
| | 0.01 | 0.1 | 200 | 0.01 | 0.1 | 200 | 0.01 | 0.1 | 200 |
| FedAvg | $62.3_{\pm 0.62}$ | $79.2_{\pm 0.53}$ | $92.1_{\pm 0.14}$ | $36.7_{\pm 0.45}$ | $48.2_{\pm 0.29}$ | $60.5_{\pm 0.11}$ | $29.4_{\pm 0.23}$ | $36.6_{\pm 0.17}$ | $49.5_{\pm 0.19}$ |
| FedProx | $70.8_{\pm 0.34}$ | $83.0_{\pm 0.19}$ | $93.2_{\pm 0.10}$ | $36.9_{\pm 0.18}$ | $48.7_{\pm 0.21}$ | $67.0_{\pm 0.17}$ | $30.1_{\pm 0.19}$ | $\mathbf{36.9}_{\pm 0.23}$ | $50.20_{\pm 0.10}$ |
| MOON | $65.4_{\pm 0.25}$ | $77.7_{\pm 0.31}$ | $\mathbf{94.1}_{\pm 0.18}$ | $37.2_{\pm 0.28}$ | $49.1_{\pm 0.25}$ | $\mathbf{67.6}_{\pm 0.16}$ | $30.7_{\pm 0.26}$ | $36.7_{\pm 0.14}$ | $\mathbf{50.6}_{\pm 0.18}$ |
| FSL | $54.1_{\pm 0.20}$ | $66.7_{\pm 0.31}$ | $77.0_{\pm 0.16}$ | $30.5_{\pm 0.15}$ | $29.4_{\pm 0.12}$ | $37.6_{\pm 0.08}$ | $19.8_{\pm 0.33}$ | $25.4_{\pm 0.29}$ | $31.0_{\pm 0.21}$ |
| FedSimCLR | $65.6_{\pm 0.29}$ | $72.1_{\pm 0.11}$ | $81.9_{\pm 0.07}$ | $32.4_{\pm 0.25}$ | $35.0_{\pm 0.24}$ | $42.3_{\pm 0.14}$ | $21.7_{\pm 0.31}$ | $24.3_{\pm 0.28}$ | $33.8_{\pm 0.25}$ |
| Orchestra | $79.6_{\pm 0.08}$ | $82.1_{\pm 0.12}$ | $81.7_{\pm 0.06}$ | $37.4_{\pm 0.13}$ | $40.3_{\pm 0.09}$ | $40.9_{\pm 0.07}$ | $26.2_{\pm 0.07}$ | $28.8_{\pm 0.08}$ | $29.7_{\pm 0.10}$ |
| FedEMA | $78.8_{\pm 0.16}$ | $81.9_{\pm 0.09}$ | $84.1_{\pm 0.06}$ | $37.2_{\pm 0.09}$ | $40.3_{\pm 0.07}$ | $37.6_{\pm 0.04}$ | $27.0_{\pm 0.16}$ | $29.1_{\pm 0.13}$ | $38.9_{\pm 0.15}$ |
| FedU | $74.7_{\pm 0.38}$ | $77.6_{\pm 0.30}$ | $83.2_{\pm 0.18}$ | $32.1_{\pm 0.21}$ | $36.7_{\pm 0.19}$ | $39.8_{\pm 0.13}$ | $24.6_{\pm 0.18}$ | $26.3_{\pm 0.28}$ | $30.1_{\pm 0.12}$ |
| FeatARC | $83.3_{\pm 0.34}$ | $\mathbf{84.6}_{\pm 0.19}$ | $86.7_{\pm 0.20}$ | $42.5_{\pm 0.19}$ | $\mathbf{54.6}_{\pm 0.19}$ | $67.3_{\pm 0.06}$ | $31.6_{\pm 0.31}$ | $32.9_{\pm 0.24}$ | $37.5_{\pm 0.26}$ |
| MFCL(BN) | $15.3_{\pm 0.41}$ | $32.5_{\pm 0.33}$ | $86.4_{\pm 0.12}$ | $14.1_{\pm 0.37}$ | $28.3_{\pm 0.38}$ | $51.3_{\pm 0.31}$ | $12.9_{\pm 0.28}$ | $22.1_{\pm 0.26}$ | $29.5_{\pm 0.19}$ |
| MFCL(GN) | $79.3_{\pm 0.14}$ | $79.8_{\pm 0.12}$ | $82.6_{\pm 0.11}$ | $41.5_{\pm 0.21}$ | $43.2_{\pm 0.16}$ | $47.1_{\pm 0.20}$ | $29.6_{\pm 0.13}$ | $30.4_{\pm 0.12}$ | $38.3_{\pm 0.12}$ |
| MFCL(TN) | $\mathbf{84.1}_{\pm 0.22}$ | $\mathbf{84.6}_{\pm 0.21}$ | $85.3_{\pm 0.19}$ | $\mathbf{44.4}_{\pm 0.17}$ | $45.5_{\pm 0.21}$ | $47.8_{\pm 0.13}$ | $\mathbf{33.1}_{\pm 0.30}$ | $33.9_{\pm 0.14}$ | $42.2_{\pm 0.18}$ |

## 5.2 Baselines

We consider the state-of-the-art works in self-supervised FL (SSFL) and supervised FL (SFL) that are proposed to address data heterogeneity. In the field of SSFL, we compare MFCL with FedU Zhuang et al. (2021) and FedEMA Zhuang et al. (2022).[6] SimCLR Chen et al. (2020) with FedAvg as another baseline, denoted as FedSimCLR. We also consider FeatArc Wang et al. (2023a), a recent approach in decentralized learning that aims to address the challenge of learning from decentralized, heterogeneous data using contrastive learning. FedAvg McMahan et al. (2016), FedProx Li et al. (2020), and MOON Li et al. (2021) are fully supervised FL. FSL Han et al. (2021) is our baseline for federated split learning.

There are two main baselines for implementing SFL in resource-limited clients: HeteroFL Diao et al. (2020) and Split-MIX Hong et al. (2022). Those baselines and the majority of current resource-compatible schemes are in the SFL field, where the number of FL rounds and the computation burden of local updates are completely different from SSFL schemes. For a fair comparison, we report the communication burden, computation burden, and memory footprint from those papers.

## 5.3 Accuracy Performance

We train CIFAR-10 over $M = 10$ clients on ResNet-18, CIFAR-100 over $M = 100$ clients on ResNet-18, and Tiny-ImageNet over $M = 100$ clients on ResNet-50. We consider three heterogeneous and CIB scenarios ($\xi = 0.1$ and $0.01$), and a homogeneous and CB scenario ($\xi = 200$). Each scenario is run 3 times, and the mean and standard deviation of the top-1 accuracy on a *uniform test set* is reported.[7]

**Improved accuracy in heterogeneous and CIB scenarios**. From the numerical results in Table 2, one can see that, in severe heterogeneous and CIB datasets, MFCL(TN) significantly outperforms other SFL and SSFL methods. Furthermore, MFCL(TN) outperforms MFCL(BN) and MFCL(GN) in those scenarios. The first reason is that MFCL benefits from self-supervised learning, which is mathematically and experimentally proved to be more robust to CIB data Liu et al. (2021). In supervised learning, the expected values of the squares of the lengths of the gradient vectors corresponding to different classes are approximately related to the square of the number of samples in those classes Anand et al. (1993); Yu et al. (2020c). This means that in supervised learning when the dataset is CIB, the gradients of the classes with more samples are larger

---

[6]The code for these baselines is not publicly available. We implemented them with the parameters reported in their papers.

[7]Unless otherwise specified, we assume that all clients participate in all rounds (i.e., $Q = M$).

Table 3: Communication burden $\alpha$, computational complexity $\beta$, and memory $\gamma$ in FedEMA and MFCL. ResNet-18 is trained on CIFAR-10 with $M = 10$ and $\xi = 0.1$. ResNet-50 is trained on Tiny-ImageNet with $M = 100$ and $\xi = 0.1$. Results are reported per client.

| | | | ResNet-18 | | | | |
|---|---|---|---|---|---|---|---|
| | #bits (p1) | FL rounds | Rep. size | #bits (p2) | $\alpha$ (bits) | $\beta$ (FLOPs) | $\gamma$ (Byte) |
| FedEMA | $21\times10^8$ | 100 | 0 | **0** | $\mathbf{21\times10^{10}}$ | $72\times10^{13}$ | $63\times10^8$ |
| FedSimCLR | $21\times10^8$ | 800 | 0 | **0** | $17\times10^{11}$ | $69\times10^{12}$ | $31\times10^8$ |
| MFCL | $\mathbf{11\times10^6}$ | 15 | $26\times10^8$ | $25\times10^{10}$ | $25\times10^{10}$ | $\mathbf{53\times10^9}$ | $\mathbf{11\times10^7}$ |

| | | | ResNet-50 | | | | |
|---|---|---|---|---|---|---|---|
| | #bits (p1) | FL rounds | Rep. size | #bits (p2) | $\alpha$ (bits) | $\beta$ (FLOPs) | $\gamma$ (Byte) |
| FedEMA | $45\times10^8$ | 100 | 0 | **0** | $44\times10^{10}$ | $33\times10^{14}$ | $17\times10^9$ |
| FedSimCLR | $45\times10^8$ | 800 | 0 | **0** | $36\times10^{11}$ | $20\times10^{13}$ | $62\times10^8$ |
| MFCL | $\mathbf{23\times10^6}$ | 15 | $52\times10^8$ | $50\times10^9$ | $\mathbf{50\times10^9}$ | $\mathbf{70\times10^{10}}$ | $\mathbf{23\times10^7}$ |

‘p’, ‘#’, ‘Rep.’ stand for ‘phase’, ‘Number of’, and ‘Representations’, respectively.

than those of the classes with fewer samples, which leads to the gradients being biased toward the class with more samples. In contrast to this, self-supervised learning does not rely on the class labels to calculate the loss, and thus, the gradients are less biased, and the model can better learn the classes with fewer samples. Therefore, with self-supervised learning, we can mitigate the *bias of labels*.

Second, MFCL benefits from TN. In severe heterogeneous and CIB scenarios, in addition to the bias of labels, we face what we call the *bias of features*. The bias of labels can be mitigated by new designs of the network (e.g., MFCL instead of E2E learning). However, the bias of features is the natural result of having clients with heterogeneous CIB data. One effective way to moderate the bias of features is to use properly designed normalization techniques. TN mitigates the bias of features when data is heterogeneous and CIB.[8]

**Accuracy in homogeneous and CB scenarios**. MFCL works best when datasets are heterogeneous and CIB, which is a practical scenario. However, in the homogeneous and CB scenarios, SFL methods outperform SSFL and MFCL. The reasons are as follows. First, when clients have homogeneous CB data, the labels are homogeneous, and the gradients are not biased. In those scenarios, SFL works better than SSFL Wu & He (2018). Second, when the mini-batch is CB and large enough, normalizing the data over the entire mini-batch (i.e., BN) works better than the sample-based normalization (e.g., GN or TN). The reason is that the batch statistics estimated from a large CB mini-batch are more accurate, yielding smaller variations of statistics from one mini-batch to another, and thus, it leads to better generalization performance.

### 5.4 Client Side Resource Saving by MFCL

**Communication burden.** In FL, the complexity of communication channels refers to the data traffic and the frequency of communication. In the following, we compare the communication complexities of MFCL and the traditional E2E SSFL methods in both aspects.

*- Data traffic*: Traditional E2E SSFL methods require sending a huge number of model parameters to the server in each FL round. In MFCL, clients only train a shallow module federally to learn the low-level features, which requires only a few FL rounds. Therefore, the data traffic of training phase 1 in MFCL is far lighter than that of the E2E SSFL methods. Meanwhile, MFCL has additional data traffic in training phase 2, which does not exist in the traditional E2E SSFL settings. Based on the simulations, considering a realistic scenario (e.g., using single precision format (32 bits) and adopting an error correction code with code rate $\frac{1}{3}$, which is used in all real-world communication systems) to transmit the model parameters in training phase 1 and representation vectors in training phase 2, we found that the overall data traffic of MFCL is lower than the corresponding E2E methods. Table 3 compares the communication burden of MFCL with that of FedSimCLR. In training phase 1, the communication burden, $\alpha_1$, of MFCL is computed by

---

[8]The advantage of TN is also demonstrated by t-SNE visualization of the learned clusters, shown in Figure 9 in the Appendix.

$$\alpha_1(\text{bits}) = \left(|\boldsymbol{\theta}_t^{\text{fed}}| + |\boldsymbol{\theta}_t^m|\right) \times 32 \times R \times T, \tag{4}$$

where $|\boldsymbol{\theta}_t^{\text{fed}}|$, $|\boldsymbol{\theta}_t^m|$, $R = \frac{1}{3}$, and $T$ denote the number of model parameters in the downlink, number of model parameters in the uplink, channel code rate, and number of FL rounds, respectively. In training phase 2, the communication burden, $\alpha_2$, of MFCL is computed by:

$$\alpha_2(\text{bits}) = (|\boldsymbol{z}^m| \times \rho) \times 32 \times R, \tag{5}$$

where, $|\boldsymbol{z}^m|$ is the size of each representation vector from client $m$ and $\rho$ is the number of representation vectors. The size of one representation vector depends on the kernel size and stride of the last layer of the client module, and the number of data samples in the clients. For example, in ResNet-18, the size of one representation vector is $|\boldsymbol{z}^m| = 32 \times 32 \times 64 \times |D^m|$, where the stride is 1 and the number of filters is 64. For CIFAR-10 over $M = 10$ clients, $|D^m| = 5000$; and for Tiny-ImageNet over $M = 100$ clients, $|D^m| = 1000$. Each representation vector contains 2 different augmented versions of each data sample. In MFCL, each client sends $\rho = 8$ representation vectors to the rServer (i.e., 16 different augmented samples for each data sample). The total communication burden for MFCL is $\alpha = \alpha_1 + \alpha_2$ which is reported in Table 3. Clearly, with more data in clients, the data traffic of MFCL would increase linearly. However, more data might need larger models in E2E SSFL, which leads to higher data traffic for FedSimCLR.[9]

- *Frequency of communications*: Each communication event incurs overhead costs, including channel setup, data encoding/decoding, handoff management[10] Kukliński et al. (2014), and session management Hasan et al. (2019). High-frequent communication increases latency and network load, potentially leading to delays and congestion, especially in bandwidth-limited or high-latency networks like mobile networks. Frequent communication also drains the battery life of mobile devices and raises operational costs for servers and infrastructure Mao et al. (2017). As reported in Table 3, each client utilizing MFCL uses the communication channels 16 times (i.e., 15 FL rounds plus one time to send the representation vectors), which is approximately 6 times fewer than FedEMA and 50 times fewer than FedSimCLR. This demonstrates that MFCL significantly outperforms other E2E SSFL methods in terms of communication burden.

**Memory and computational efficiency.** On the side of clients, the memory footprint to store model parameters, feature maps, and gradients, is an important issue. The memory footprint $\gamma$ of each client is computed by

$$\gamma\,(\text{Bytes}) = \left(|\boldsymbol{\theta}^m| + |\boldsymbol{g}^m| + \sum_l |\boldsymbol{w}_l{}^m|\right) \times 2K \times 4, \tag{6}$$

where $|\boldsymbol{\theta}^m|$, $|\boldsymbol{g}^m|$, $2K$, and $|\boldsymbol{w}_l{}^m|$ are the number of model parameters, the number of gradients, the size of $B^m$, and the size of feature maps of different layers of the network, respectively. The feature map size in each layer depends on the kernel size, the stride and the number of filters in the layer. As shown in Table 3, the memory footprint of MFCL on the side of clients is much lower than the FedSimCLR. MFCL also has lower complexity in terms of FLOPs, which makes it suitable for resource-limited clients.

**MFCL and other resource-compatible baselines.** In Table 4, the accuracy, communication burden $\alpha$, computation burden $\beta$, and memory footprint $\gamma$ are reported for MFCL(TN) and other schemes designed for resource-limited clients in SFL. FedPAC Xu et al. (2023) is another promising baseline to reduce the communication burden and address the data heterogeneity. Instead of updating all clients' classifiers simultaneously, FedPAC shares only classifiers that are relevant to a given client, minimizing communication overhead. However, FedPAC requires more local updates, which increases the computational burden and memory footprint on the client side, making it less suitable for resource-limited clients.

One can see that MFCL(TN) has better accuracy in severe heterogeneous data scenarios while reducing the client memory footprint, with an increase in communication burden.

---

[9]Another important aspect for MFCL is that not all clients who participated in training phase 1 necessarily participate in training phase 2. Figure 8 in the Appendix shows that MFCL is robustly working even in that situation.

[10]Mobile devices may move between different wireless network coverages, called the cells, requiring handoffs that involve signaling and maintaining session continuity. Frequent communication increases the number of handoffs.

Table 4: $\alpha$, $\beta$, and $\gamma$ in MFCL(TN) and resource compatible baselines on ResNet-18 ($M = 10$).

| Method | ACC | | $\alpha$(bits) | $\beta$(FLOPs) | $\gamma$(Byte) |
|---|---|---|---|---|---|
| | $\xi=0.01$ | $\xi=200$ | | | |
| HeteroFL | 49.3 | 88.6 | $15\times10^{12}$ | $21\times10^{11}$ | $55\times10^{8}$ |
| Split-Mix | 50.7 | 88.2 | $45\times10^{11}$ | $\mathbf{27\times10^{9}}$ | $70\times10^{7}$ |
| FedPAC | 61.7 | 80.3 | $\mathbf{12\times10^{10}}$ | $44\times10^{11}$ | $18\times10^{10}$ |
| MFCL(TN) | **84.1** | 84.5 | $25\times10^{10}$ | $53\times10^{9}$ | $\mathbf{11\times10^{7}}$ |

Table 5: TN in FedSimCLR with ResNet-18 (CIFAR-10, $M$=10).

| | $\xi = 0.01$ | | $\xi = 0.1$ | |
|---|---|---|---|---|
| | FedSimCLR | MFCL | FedSimCLR | MFCL |
| BN | 65.6 | 15.3 | 72.1 | 32.5 |
| GN | 68.1 | 79.3 | 74.7 | 79.8 |
| TN | **72.3** | **84.1** | **78.2** | **84.5** |

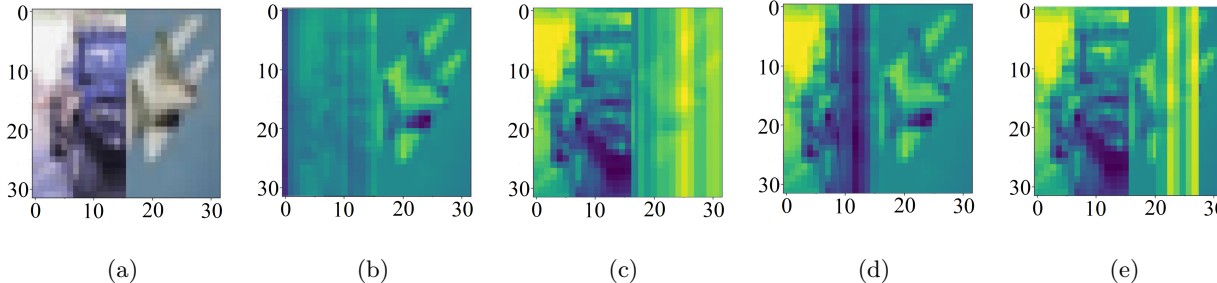

(a)  (b)  (c)  (d)  (e)

Figure 4: Visualization of features learned by MFCL(BN) and MFCL(TN) with 2 clients. Client 1: 4950 airplanes, 50 cars. Client 2: 4950 cars, 50 airplanes. (a) The synthesized test image with the left showing a car and the right showing an airplane. The saliency map of gradients for (b) client 1 with MFCL(BN), (c) client 2 with MFCL(BN), (d) client 1 with MFCL(TN), and (e) client 2 with MFCL(TN).

### 5.5 More Simulation Results for TN

**Addressing data heterogeneity with TN.** Figure 4 shows that TN has better performance over BN in learning classes with a smaller number of samples. Let us consider 2 clients each with 5000 samples. Client 1 has 4950 'airplanes' and 50 'cars'. Client 2 has 4950 'cars' and 50 'airplanes'. MFCL(BN) and MFCL(TN) are used to train ResNet-18 on those clients. When training is done, we use the synthesized samples to see the gradients' saliency map. Figure 4a depicts a synthesized sample with half showing a car and half showing an airplane. Figures 4b and 4c show the gradients' saliency map for MFCL(BN) on clients 1 and 2, respectively. For client 1, 'cars' are rare and the network does not learn good features for them, while client 2 struggles with rare 'airplanes'. Figures 4d and 4e show the gradients' saliency map for MFCL(TN) on the same clients. Clearly, MFCL(TN) has learned better features for both classes.

**TN in other SSFL methods.** To see if TN is still effective in other contrastive FL settings, we applied TN (instead of BN) to FedSimCLR after the first two Conv2D layers and results are presented in Table 5. One can see that (i) TN can improve the performance of FedSimCLR and (ii) MFCL(TN) outperforms FedSimCLR (with BN, GN, or TN) when clients have heterogeneous and CIB data.

One of the advantages of MFCL is that the batch size in the client module does not need to be large. The effects of other hyperparameters related to TN, such as batch size and number of channels per group, $\Lambda$, are detailed in Tables 16 and 17 in the Appendix.

### 5.6 Extra Privacy

The general assumption for preserving privacy in FL is that the clients are honest, and the server is honest but curious Nasr et al. (2018); Le et al. (2023). MFCL has two servers: gServer for model parameter transactions and rServer for training on the received representations.

Table 6: The accuracy performance of MFCL with additional privacy provided by SA, DM and both SA and DM. ResNet-18 is trained on CIFAR-10 with $M = 10$ clients and $\xi = 0.01$.

|          | SA   | DM   | SA+DM | No Encryption |
|----------|------|------|-------|---------------|
| MFCL(TN) | 84.1 | 82.4 | 82.4  | 84.1          |

Table 7: Accuracy comparison between FedEMA and MFCL(TN) in the data setting of FedEMA Zhuang et al. (2022) with $M = 5$. For CIFAR-10, each client has 2 classes. For CIFAR-100, each client has 20 classes.

|          | CIFAR-10 | CIFAR-100 |
|----------|----------|-----------|
| FedEMA   | 83.3     | 61.7      |
| MFCL(TN) | **85.6** | **61.9**  |

### 5.6.1 Extra privacy-preserving mechanisms in training phase 1 and 2

**Training phase 1.** Training phase 1 of MFCL follows the traditional FL. Any privacy-preserving algorithms such as Differential Privacy Abadi et al. (2016), Secure Aggregation (SA) Bonawitz et al. (2017), or Homomorphic Encryption Zhang et al. (2020a) can be applied to the model parameters. We adopted SA in Table 6 to avoid extra computation burden in clients.

**Training phase 2.** Since rServer does not have access to the client module's model parameters, reconstructing data from harshly augmented samples using a random decoder is challenging. We also provide extra privacy by applying DataMix Liu et al. (2020b) to MFCL, and the results are listed in Table 6.[11]

### 5.6.2 Possible Attack Scenarios

We analyzed different scenarios, considering whether the gServer implements SA, whether the rServer applies DM, and whether the two servers are colluding. SA is adopted to preserve the privacy of the client's data at the gServer. Note that if two servers collude, SA cannot provide extra privacy at the rServer.[12] The goal is to reconstruct the original sample **x** from the representations of augmented data with or without DataMix. The simulation results are shown in Table **??**.

**Case 1: C1D0 (Servers collude and there is no DataMix)** In this scenario, the gServer transmits the averaged gradients to the rServer. The rServer, having access to both the gradients and the representations of the augmented data, is capable of reconstructing the augmented data samples. However, a more complex challenge lies in reconstructing the original data samples from the augmented data, without direct access to the original data or labels necessary to train a network for such reconstruction.

**Case 2: C1D1 (Servers collude and DataMix is applied at clients' side)** This case is similar to case 1, except that the clients use DataMix. One can see the reconstruction is even more challenging.

**Case 3: C0D0 (Servers do not collude and there is no DataMix)** In this case, the rServer does not have the gradients. Therefore the decoding process to reconstruct the augmented data from the representations of the augmented data should start from pure noisy gradients. Again, the more challenging step is to reconstruct the main data from the augmented data, without having access to data samples or the labels.

**Case 4: C0D1 (Servers do not collude and DataMix is applied at clients' side)** This case is similar to the case 3, except that the clients use DataMix. One can see the reconstruction is not possible.

### 5.7 Detailed Comparison with FedEMA

From Table 2 one can see MFCL(TN) works better than FedEMA Zhuang et al. (2022) which is the state-of-the-art baseline for SSFL. In what follows, we compare MFCL with FedEMA from various aspects:

---

[11]It is also possible to add more layers to client modules, which creates a trade-off between privacy and accuracy (Table 1).

[12]We implemented the basic form of SA, where gradients are masked on the client side so that the masks cancel out during aggregation. This way, the gServer does not have access to the individual client gradients but can still access the aggregated gradients. Alternative methods, such as Secure Multi-Party Computation, or combinations of SA with Homomorphic Encryption or Differential Privacy, can ensure that even the aggregated gradients remain inaccessible to the gServer. Additionally, employing a Trusted Execution Environment (TEE) at the gServer could further enhance privacy without extra computation burden on the side of clients.

Table 8: Attack scenarios.

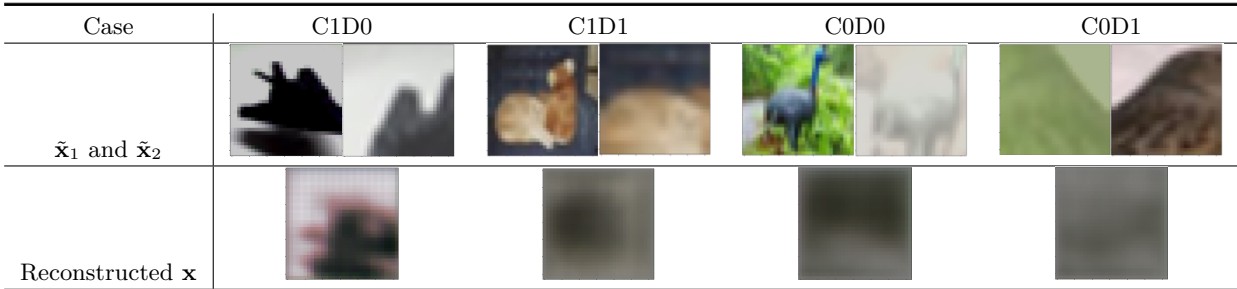

| Case | C1D0 | C1D1 | C0D0 | C0D1 |
|------|------|------|------|------|
| $\tilde{\mathbf{x}}_1$ and $\tilde{\mathbf{x}}_2$ | | | | |
| Reconstructed $\mathbf{x}$ | | | | |

**CB dataset in each client.** In FedEMA, the client's dataset is assumed to be CB (e.g., each client has 2 classes of CIFAR-10). In Table 2, we utilized the Dirichlet distribution. However, for fair comparison in Table 7, we implemented MFCL(TN) using the data distribution from FedEMA with 5 clients (i.e., 2 classes per client for CIFAR-10 and 20 classes per client for CIFAR-100). One can see that with 5 clients and CB datasets, the accuracy performance of FedEMA is close to that of MFCL(TN).

**Number of clients.** As reported in Table 4 of FedEMA Zhuang et al. (2022), the accuracy performance of FedEMA declines with increasing the number of clients, rendering it unsuitable for cross-device FL. MFCL works well with 100 clients (Table 2) or even more, which is more realistic in cross-device scenarios.

**Number of samples per client.** As reported in Table 12 of Zhuang et al. (2022), the top 1 accuracy of FedEMA Zhuang et al. (2022) improves with increasing the number of samples per client. The reason is that in FedEMA a deep network, such as ResNet-50, is trained federally, and having more samples in each client reduces the likelihood of overfitting and enhances the training performance. In contrast, in MFCL a shallow autoencoder is trained federally, making it feasible to work effectively with a smaller number of data samples, which is more realistic in cross-device FL.

**Communication burden and memory footprint.** Table 3 presents a comparison of the communication burden, computation burden, and memory footprint between FedEMA and MFCL(TN). The communication burden of FedEMA is marginally lower than that of MFCL(TN) and FedSimCLR. However, FedEMA incurs a substantially higher computation burden and memory footprint. The reason is that FedEMA needs to train and store two networks (i.e., online and target networks) at each client, leading to higher computational demands and memory usage.

FedEMA works perfectly well in cross-silo applications where (1) the number of clients is not large and (2) each client has enough data samples and resources. MFCL(TN) is designed for cross-device applications where (1) the number of clients can be large, and (2) clients do not have huge data, communication, computation, and memory resources.

## 6 Conclusion

**To summarize.** We have developed MFCL, as a novel approach to FL, transitioning from an E2E learning to a modular framework to leverage contrastive learning in resource-constrained clients. We also proposed Twin Normalization which is effective in addressing the challenges of heterogeneous CIB data at clients. Our simulations demonstrate that MFCL(TN) achieves superior and more stable performance in highly heterogeneous and CIB data scenarios. Furthermore, MFCL exhibits reduced communication burden, memory footprint, and computational complexity on the client side. It is noteworthy that MFCL works effectively with group normalization, indicating that it performs well even without twin normalization.

**Limitations.** MFCL has great potential as a resource-effective approach for cross-device FL and can handle severe heterogeneous and CIB datasets. However, there exist certain limitations for MFCL. When the number of data samples per client is huge (e.g., in cross-silo FL), the client module should include additional layers to maintain performance. This requirement can compromise the resource efficiency advantages of MFCL.

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

# A    Related Works

## A.1    Supervised FL with Heterogeneous and CIB Datasets

There are plenty of works in SFL aiming to address the issue of data heterogeneity among the clients. The first approach is to leverage a shared public or synthesized dataset on clients and/or the server to find a solution for the local models that under-represent the patterns at the clients Zhao et al. (2018); Hao et al. (2021). The second approach is based on regularization. FedProx Li et al. (2020) regularizes the Euclidean distance between the local and global models. MOON Li et al. (2021) uses contrastive loss to maximize the agreement of the representations learned by the local and global models. SCAFFOLD Karimireddy et al. (2020) reduces the bias in the gradients by introducing some control variates. FedDyn Acar et al. (2021) dynamically changes the local objectives at each FL round to ensure that the local optimum is consistent with the global optimum. The third approach is based on reducing the bias of the aggregated parameters at the server. The authors in Hsu et al. (2019), leveraging the momentum update on the server, tried to alleviate the oscillations resulting from averaging the biased gradients. The authors in Wang et al. (2020a) proposed to match the local updates while aggregating to reduce the effect of heterogeneous data. In particular, to achieve a fair (if not perfect) performance of the model on the local datasets, researchers have recently studied personalized FL, through various approaches, e.g., by treating each client as a task in a multi-task learning framework Smith et al. (2017) or by dividing clients into different clusters based on their tasks and performing cluster-based aggregation as in Sattler et al. (2020); Ghosh et al. (2020).

In addition to the issue of data heterogeneity, the number of data samples for each class in a client's dataset is not necessarily the same. This means that each client's data might be often CIB rather than CB. Over the past few years, some efforts have been devoted to addressing the issue of CIB data in the clients by regularization techniques as in Wang et al. (2021a), data level approaches such as data augmentation in Duan et al. (2020), or data distribution estimation in Yang et al. (2021). However, most of the existing FL methods, including all the algorithms mentioned above, assume that clients have fully labelled data. Figure 6 shows different scenarios of data distribution of clients in FL.

## A.2    Contrastive Learning and Self-Supervised FL

Researchers have studied contrastive and non-contrastive approaches for SSFL Zhuang et al. (2022); Zhang et al. (2020b); Miao & Koyuncu (2022); Shi et al. (2021); Yu et al. (2020a). The contrastive loss, used in Oord et al. (2018), is defined between two augmented views $(i, j)$ of the same data sample in a mini-batch of the size of $2K$, as follows:

$$\mathscr{L} = -\frac{1}{2K} \sum_{i,j \in B} \log \frac{\exp\left(\frac{\text{sim}(\boldsymbol{h}_i, \boldsymbol{h}_j)}{\tau}\right)}{\sum_{k=1}^{2K} \mathbb{1}_{[k \neq i]} \exp\left(\frac{\text{sim}(\boldsymbol{h}_i, \boldsymbol{h}_k)}{\tau}\right)}, \tag{7}$$

where $\boldsymbol{h}_i$ and $\boldsymbol{h}_j$ are the hidden representations of two positive examples[13], $\mathbb{1}_{[k \neq i]}$ is an indicator function that is equal to 1 if $k \neq i$, $\text{sim}(\boldsymbol{h}_i, \boldsymbol{h}_j)$ is the cosine similarity between two vectors of the hidden representations, $\tau$ is a temperature scalar, and $B$ is a randomly sampled mini-batch consisting of augmented pairs of images. For representation learning without a contrastive loss, the authors in Lu et al. (2022) proposed to use prior information about the client's dataset, instead of labels, to apply supervised learning methods to unlabeled data. Orchestra Lubana et al. (2022) is proposed as an SSFL scheme to partition the client's data into distinguishable clusters. Recently, the authors in Zhuang et al. (2022) proposed a divergence-aware aggregation for FL, based on different self-supervised learning methods. Unfortunately, their scheme does not perform well when the number of clients is large (e.g., more than 10). Also, they did not consider the case in which the clients' datasets are CIB.

---

[13]By "examples", we mean augmented data samples. Positive examples are two differently augmented versions of the same image, and negative examples are the augmented versions of other images.

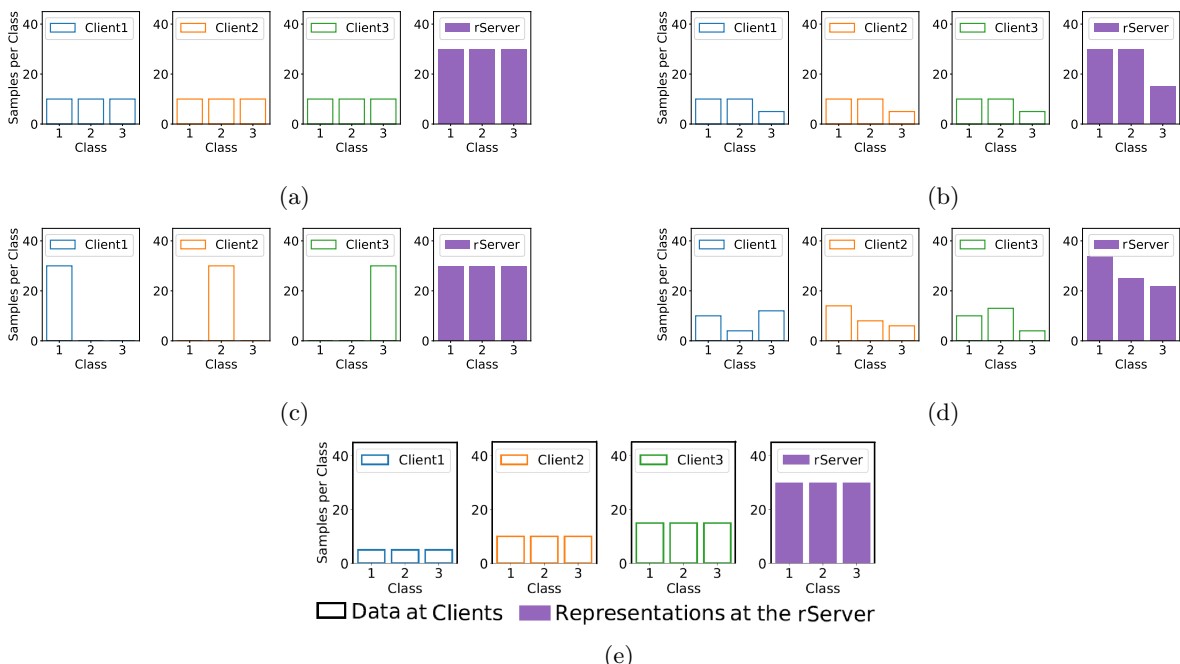

Figure 6: Different scenarios for the (raw) data at the clients and the contrastive rServer representation mini-batches (simply, representations) at the rServer. (a) clients have homogeneous and CB data; and the representations are CB at the rServer, (b) clients have homogeneous and CIB data; and the representations are CIB at the rServer, (c) clients have heterogeneous and CIB data; and the representations are CB at the rServer, (d) clients have heterogeneous and CIB data; and the representations are CIB at the rServer, and (e) clients have heterogeneous and CB data; and the representations are CB at the rServer.

Table 9: The architecture of the client module and its decoder on ResNet-18. For the convolutional layer (Conv2D) and transposed convolution layer (Conv2DTranspose), we present the list of the parameters in the sequence of input and output dimensions, kernel size, stride, and padding. For the max pooling layer (MaxPool2D), we show the list of the kernel and stride.

| Layer | Details |
|---|---|
| 1 | Conv2D(3, 64, 3, 1, 1) |
| 2 | TN, ReLU |
| 3 | Conv2D(64, 64, 3, 1, 1) |
| 4 | TN, ReLU, MaxPool2D(2, 2) |
| 5 | Conv2DTranspose(64, 64, 3, 1, 1) |
| 6 | TN, ReLU |
| 7 | Conv2DTranspose(64, 3, 3, 1, 1) |
| 8 | TN, ReLU |
| 9 | Conv2D(3, 3, 3, 1, 1) |

# B   Experimental Details

In this section, we illustrate the details of the model architectures and some additional simulation scenarios.

## B.1   Heterogeneous and CIB Data

In our proposed MFCL, the data representations sent from the clients are collected at the rServer, and the contrastive rServer mini-batches are constructed with the received representation vectors at the rServer.

Table 10: Same as Table 6 for ResNet-50.

| Layer | Details |
|-------|---------|
| 1 | Conv2D(3, 64, 3, 1, 1) |
| 2 | TN, ReLU |
| 3 | Conv2D(64, 64, 3, 1, 1) |
| 4 | TN, ReLU |
| 5 | Conv2D(64, 128, 3, 1, 1) |
| 6 | TN, ReLU |
| 7 | Conv2D(128, 128, 3, 1, 1) |
| 8 | TN, ReLU, MaxPool2D(2, 2) |
| 9 | Conv2DTranspose(128, 128, 3, 1, 1) |
| 10 | TN, ReLU |
| 11 | Conv2DTranspose(128, 64, 3, 1, 1) |
| 12 | TN, ReLU |
| 13 | Conv2DTranspose(64, 64, 3, 1, 1) |
| 14 | TN, ReLU |
| 15 | Conv2DTranspose(64, 3, 3, 1, 1) |
| 16 | TN, ReLU |
| 17 | Conv2D(3, 3, 3, 1, 1) |

Therefore, we need to define the concept of CB or CIB representations at the rServer. For clarification, we provide Figure 6, which consists of data distribution in the clients and the distribution of the representations at the rServer. In Figure 6a, the clients have homogeneous and CB data in each client; In Figure 6b, the clients have homogeneous and CIB data in each client; In Figures 6c and 6d, the clients have heterogeneous and CIB data in each client; In Figure 6e, the clients have heterogeneous and CB data in each client. To sum it up, if the dataset is CB in all clients, the rServer mini-batches are always CB, even if some clients are not successful in sending their representations vectors (Figures 6a and 6e). Otherwise, the rServer mini-batches can be CB or CIB: the rServer mini-batches are CIB in Figures 6b and 6d, whereas they are CB in Figure 6c.

## B.2 Details of the Client and rServer Modules

### B.2.1 Data Distribution

Following baselines Lubana et al. (2022); Luo et al. (2021); Makhija et al. (2022), we chose M∈{10, 100} and $\xi$∈{0.01, 0.1, 0.5}. For CIFAR-10, $\xi$=0.01 implies each client has two very unbalanced classes and $\xi$=0.1 implies each client has 4 or 5 unbalanced classes, and so on. The smaller $\xi$, the more unbalanced the classes (Figure 7a), which is most realistic for FL.

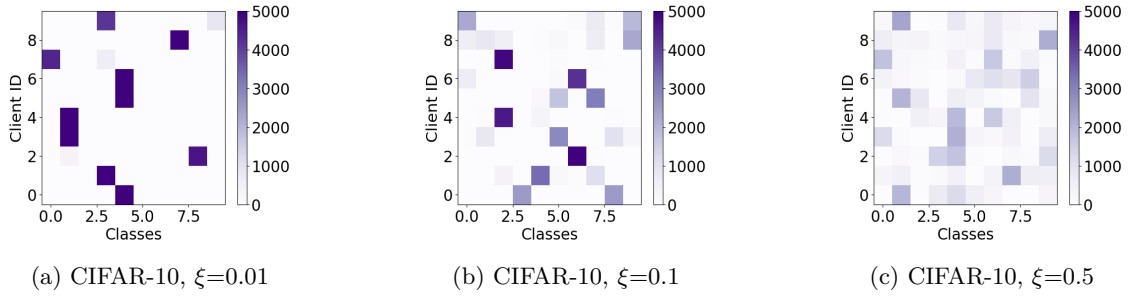

(a) CIFAR-10, $\xi$=0.01  (b) CIFAR-10, $\xi$=0.1  (c) CIFAR-10, $\xi$=0.5

Figure 7: Client data distribution with CIFAR-10 considering various values for $\xi$.

Table 11: Projection layer architecture (adopted in both ResNet-18 and ResNet-50).

| Layer | Details |
|---|---|
| 1 | Dense |
| 2 | BN, ReLU |
| 3 | Dense |
| 4 | BN, ReLU |
| 5 | Dense |
| 6 | BN |

Table 12: List of hyperparameters.

| Data | CIFAR-10/100 | Tiny-ImageNet |
|---|---|---|
| Model | ResNet-18 | ResNet-50 |
| Client module (CM) | First 2 layers | First 4 layers |
| CM batch size | 64 | 64 |
| CM optimizer | Adam | Adam |
| CM learning rate | 0.05 | 0.001 |
| FL rounds | 15 | 15 |
| rServer module (rSM) epochs | 150 | 200 |
| rSM optimizer | LARS | LARS |
| rSM learning rate | cosine, $lr_0 = 1.0$ | cosine, $lr_0 = 1.0$ |

Table 13: List of augmentation techniques.

| Dataset / Augmentation | CIFAR-10/100 | Tiny-ImageNet |
|---|---|---|
| Brightness | ✓ | ✓ |
| Contrast | ✓ | ✓ |
| Colorjitter | ✓ | ✓ |
| Grayscale | ✓ | ✓ |
| Saturation | ✓ | ✓ |
| Hue | ✓ | ✓ |
| Crop and resize | ✓ | ✓ |
| Gaussian blur | – | ✓ |
| Flip | ✓ | ✓ |

### B.2.2 Modules' structures and hyperparameters

In our experiments, the client's module includes the initial layers of a deep network, extending up to a specified cut-off layer. The decoder for the client module is designed such that each client module, in conjunction with its decoder forms a convolutional autoencoder. Considering ResNet-18, the client module (i.e., the first two layers of the network) and its decoder are trained federally, with binary cross-entropy loss, across $M = 10$ clients for CIFAR-10 and $M = 100$ for CIFAR-100. For Tiny-ImageNet, we include the first layers of ResNet-50 as the client's module and train a deeper autoencoder federally across $M = 100$ clients. The details of the client module and decoder are provided in Tables 9 and 10 for ResNet-18 and ResNet-50, respectively.

We use non-linear projection layers (Table 11) on top of the rServer module and also one linear Dense layer is used on top of the projection layers for linear evaluation. A list of other hyperparameters and augmentation strategies are listed in Tables 12 and 13, respectively.

Table 14: Accuracy comparison of MFCL and baselines on CIFAR-10 with $M = 10$ on ResNet-18, CIFAR-100 with $M = 100$ on ResNet-18, and Tiny-ImageNet with $M = 100$ on ResNet-50. The best performance is highlighted in boldface. For SSFL methods, the top-1 accuracy with linear evaluation is reported. For MFCL, BN is used at the rServer.

| Method \ $\xi$ | CIFAR-10 | | | | CIFAR-100 | | | | Tiny-ImageNet | | | |
|---|---|---|---|---|---|---|---|---|---|---|---|---|
| | 0.01 | 0.1 | 0.5 | 200 | 0.01 | 0.1 | 0.5 | 200 | 0.01 | 0.1 | 0.5 | 200 |
| FedAvg | 62.3 | 79.2 | 84.2 | 92.1 | 36.7 | 48.2 | 55.3 | 60.5 | 29.4 | 36.6 | 45.3 | 49.5 |
| FedProx | 70.8 | 83.0 | **87.6** | 93.2 | 36.9 | 48.7 | 59.2 | 67.0 | 30.1 | **36.9** | 47.0 | 50.1 |
| MOON | 65.4 | 77.7 | 86.3 | **94.1** | 37.2 | 49.1 | 60.6 | **67.6** | 30.7 | 36.7 | **47.4** | **50.6** |
| FSL | 54.1 | 66.7 | 74.3 | 77.0 | 30.5 | 29.4 | 34.3 | 37.6 | 19.8 | 25.4 | 28.2 | 31.0 |
| FedSimCLR | 65.6 | 72.1 | 79.6 | 81.9 | 32.4 | 35.0 | 39.6 | 42.3 | 21.7 | 24.3 | 30.9 | 33.8 |
| Orchestra | 79.6 | 82.1 | 82.1 | 81.7 | 37.4 | 40.3 | 41.2 | 40.9 | 26.2 | 28.8 | 30.1 | 29.7 |
| FedEMA | 78.8 | 81.9 | 83.5 | 84.1 | 37.2 | 40.3 | 40.9 | 37.6 | 27.0 | 29.1 | 31.2 | 38.9 |
| FedU | 74.7 | 77.6 | 79.1 | 83.2 | 32.1 | 36.7 | 37.2 | 39.8 | 24.6 | 26.3 | 29.4 | 30.1 |
| FeatARC | 83.3 | **84.6** | 85.2 | 86.7 | 42.5 | **54.6** | **61.0** | 67.3 | 31.6 | 32.9 | 34.8 | 37.5 |
| MFCL(BN) | 15.3 | 32.5 | 53.8 | 86.4 | 14.1 | 28.3 | 38.2 | 51.3 | 12.9 | 22.1 | 27.8 | 29.5 |
| MFCL(GN) | 79.3 | 79.8 | 80.7 | 82.6 | 41.5 | 43.2 | 45.2 | 47.1 | 29.6 | 30.4 | 33.5 | 38.3 |
| MFCL(TN) | **84.1** | 84.5 | 84.6 | 85.3 | **44.4** | 45.5 | 46.5 | 47.8 | **33.1** | 33.9 | 36.8 | 42.2 |

Table 15: $\alpha$, $\beta$, and $\gamma$ in MFCL(TN) on VGG-16-BN ($M = 10$).

| Method | ACC | | $\alpha$(bits) | $\beta$(FLOPs) | $\gamma$(Byte) |
|---|---|---|---|---|---|
| | $\xi=0.01$ | $\xi=200$ | | | |
| FedSimCLR | 65.7 | 81.3 | $21\times10^{12}$ | $86\times10^{13}$ | $39\times10^{9}$ |
| MFCL(TN) | 82.8 | 83.5 | $25\times10^{10}$ | $53\times10^{9}$ | $11\times10^{7}$ |

### B.2.3  MFCL performance on other backbones

We provided more simulation results on more data scenarios in Table 14. Also, to thoroughly assess the performance and generalizability of MFCL, we conducted additional experiments using VGG-16-BN as another backbone architecture, which is the VGG 16-layer model with batch normalization Simonyan & Zisserman (2014). This allowed us to evaluate how MFCL performs across different network structures, ensuring that its effectiveness is not limited to the ResNet family. Similar to ResNet-18 on CIFAR-10, we select the first two layers as the encoder module and design the decoder as the mirror structure of the encoder part. We train the encoder module and its decoder federally across the clients and the next layers are trained at the Server. The simulation results are presented in Table 15.

Since the client modules of the ResNet-18 and VGG-16-BN backbones are identical, the communication, computation, and memory footprints of MFCL(TN) are the same for both backbones on the client side. However, VGG-16-BN has 138 million parameters, which is why the communication, computation, and memory footprint of E2E FedSimCLR with a VGG-16-BN backbone is higher than those with a ResNet-18 backbone.

### B.2.4  Effect of the batch size on the side of clients

As outlined in Algorithm 1, the MFCL framework has two distinct phases: the first phase involves federated training of the client module and its decoder using the gServer, while the second phase focuses on training the representation modules at the rServer through contrastive learning. One of the benefits of MFCL is that the batch size in the client module, $K$, does not need to be large. The performance accuracy of MFCL with various $K$ is provided in Table 16.

---

**Algorithm 1** Modular Federated Contrastive Learning (MFCL)

---

1: **Input:** Clients' data $\{D^m\}_{m=1}^M$.
2: **Initialize:** $gServer$ with initial model $\boldsymbol{\theta}_0$, $rServer$ with initial model $\boldsymbol{\chi}_0$.
3: **Phase 1: Training Client Modules and Decoders Federally at gServer**
4: **for** $t = 1, 2, \cdots, T$ **do**
5:     $gServer$ sends $\boldsymbol{\theta}_{t-1}^{\text{fed}}$ to a subset of selected clients $\{\pi_t\}$.
6:     **for** each client $m \in \{\pi_t\}$ **in parallel do**
7:         Client downloads $\boldsymbol{\theta}_{t-1}^{\text{fed}}$.
8:         Client creates two augmented versions $\tilde{\mathbf{x}}_{i,j}^m$, $j = 1, 2$ of $\mathbf{x}_i^m \in D^m$.
9:         Client constructs contrastive input mini-batches $B^m = \left(\tilde{\mathbf{x}}_{1,1}^m, \ldots, \tilde{\mathbf{x}}_{K,1}^m, \tilde{\mathbf{x}}_{1,2}^m, \ldots, \tilde{\mathbf{x}}_{K,2}^m\right)$.
10:         Client performs the local update on the client module and the decoder.
11:         Client sends updated parameters $\boldsymbol{\theta}_t^m$ to $gServer$.
12:     **end for**
13:     $gServer$ aggregates updated parameters: $\boldsymbol{\theta}_t^{\text{fed}} = \frac{1}{\sum_{m' \in \pi_t} |D^{m'}|} \sum_{m \in \pi_t} |D^m| \boldsymbol{\theta}_t^m$.
14: **end for**
15: **Phase 2: Training Representation Modules at rServer**
16: **for** each client $m$ **in parallel do**
17:     Client generates representations $\boldsymbol{z}_{i,j}^m = f_{[\boldsymbol{\theta}_T]_e}(\tilde{\mathbf{x}}_{i,j}^m), j = 1, 2$ from the client module.
18:     Client sends the representations to $rServer$.
19: **end for**
20: $rServer$ constructs batches $B^s$ from the received representations.
21: $rServer$ updates its parameters $\chi$ by minimizing contrastive loss.
22: **Output:** The trained rServer module is broadcast to clients to ensure a global model across clients.

---

Table 16: Effect of the client module's batch size, $K$, on the performance accuracy of MFCL(TN). ResNet-18 is trained on CIFAR-10 with 10 clients and $\xi = 0.01$.

| Method \ $K$ | 8 | 16 | 32 | 64 |
|---|---|---|---|---|
| MFCL(TN) | 83.9 | 84.0 | 84.1 | 84.1 |

### B.2.5 Effect of the client dropout in training phase 2

When training phase 1 is finished, the clients send their data representations to the rServer. In real-world scenarios, however, not all the clients who participated in training phase 1 can join training phase 2. Specifically, in training phase 2, the rServer may not successfully receive the representation vectors from some clients who participated in training phase 1. This could be due to the representation vectors being corrupted or lost during transmissions, or some clients being unable to send their data representations. In this case, our assumption that the contrastive rServer mini-batches are (closely) CB might not be valid anymore. We investigated how the performance of our proposed MFCL with BN, GN, and TN is affected by client dropout, considering the scenarios where $\delta(\%)$ of clients cannot transmit their representation vectors to the rServer during training phase 2.

In Figure 8a, we trained MCFL with BN, GN, and TN on the CIFAR-10 dataset with $\xi = 200$ when $M = 10$ clients participated in training phase 1, assuming the representation vectors from $\delta(\%)$ clients are not received by the rServer in training phase 2. We repeated the same experiments in Figure 8b with $\xi = 0.01$. From Figures 8a and 8b, one can see that both TN and GN exhibit similar behaviour when some clients are dropped during training phase 2. For $\xi = 200$, the gap between TN and BN decreases as more clients are dropped in training phase 2. Meanwhile, for $\xi = 0.01$, our proposed TN continues to be the best choice even as more clients are dropped in training phase 2.

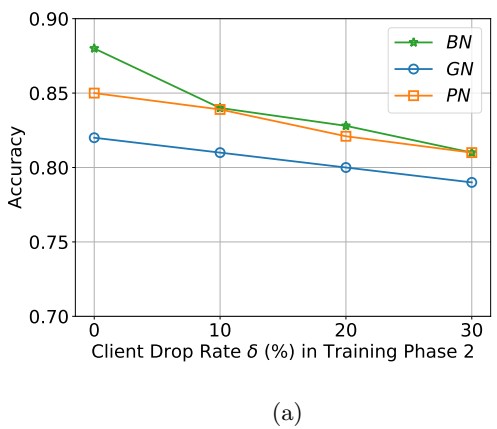 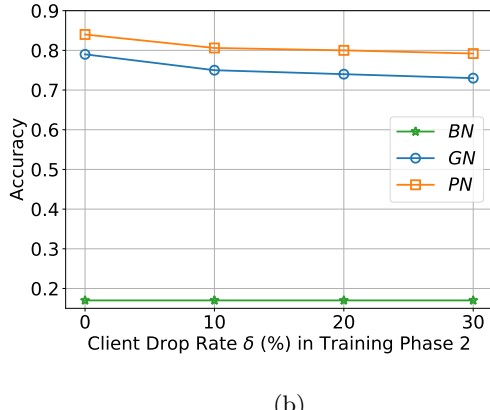

(a)                                                             (b)

Figure 8: The classification accuracy of the proposed MFCL with BN, GN, and TN, when the client dropout rate is $\delta\%$ in training phase 2. The number of clients participating in training phase 1 is $M = 10$. (a) $\xi = 200$ is considered, which is the scenario of homogeneous and CB data. (b) $\xi = 0.1$ considered, which is a severe case of data heterogeneous clients with CIB data.

Table 17: Effect of the number of channels per group on the performance accuracy of MFCL(TN) and MFCL(GN). ResNet-18 is trained on CIFAR-10 with 10 clients and $\xi = 0.1$.

| Method \ $\Lambda$ | 2 | 4 | 8 | 16 | 32 | 64 |
|---|---|---|---|---|---|---|
| MFCL(GN) | 79.2 | 79.4 | 79.5 | 79.8 | 79.8 | 79.6 |
| MFCL(TN) | 83.1 | 83.7 | **84.5** | 84.5 | 84.2 | 83.9 |

## B.3 More Simulation Results for TN

### B.3.1 Effect of the number of channels per group in TN

We evaluate the effect of the number of channels per group, $\Lambda$, in TN in Table 17. Note that in the extreme case of 1 channel per group, GN is equivalent to Instance Normalization Ulyanov et al. (2016) but TN still normalizes the two examples together based on the statistics of both channels (i.e., 1 channel per group for each example). One can compare the accuracy performance of MFCL(TN) and MFCL(GN) when the number of channels per group is the same in both methods (i.e., $\Lambda = 4$ channels per group in MFCL(GN) vs $\Lambda = 2$ channels per group for MFCL(TN), which means 4 channels in total). Even using as few as 2 channels per group for each example (i.e., 4 channels in total), MFCL(TN) has substantially better accuracy than MFCL(GN).

### B.3.2 Implementation

We implemented TN with a few lines of code in TensorFlow in Table 18. We simply need to calculate the mean and variance, aligned with the respective axes for TN.

## B.4 Visualization of learned clusters by MFCL

In this subsection, the features of the learned clusters are visualized to provide a deeper understanding of the advantages of the proposed normalization method, TN. Each client is assumed to have 2 classes of the CIFAR-10 dataset, with $M = 10$. Our objective is to evaluate the quality of the learned clusters with MFCL, at the output of the projection layers when BN, GN, or TN is utilized in the client modules and decoders.

Figure 9a presents the t-SNE visualization of raw input data at a client, while Figures 9b-9d display the t-SNE visualizations of the clusters learned from the data representations of all clients at the rServer. Figure

Table 18: Twin normalization in TensorFlow.

```python
def TwinNorm(x, gamma, beta, CP, eps=1e-5):
  # x: features with shape [N,C,H,W]
  # beta, gamma: shift and scale, with shape [1,C,1,1]
  # CP: number of channels per group

  N, C, H, W = x.shape
  G=C/CP
  x = tf.reshape(x, [N, G, C // G, H, W])
  meanGN, varGN = tf.nn.moments(x, [2, 3, 4], keep dims=True)

  meanP=tf.math.add(meanGN[0:N/2]+meanGN[N/2:N])/2
  varP=tf.math.add(varGN[0:N/2]+varGN[N/2:N])/2

  meanTN=tf.concat([meanP, meanP], axis=0)

  varTN=tf.concat([varP, varP], axis=0)

  x = (x - meanTN) / tf.sqrt(varTN + eps)
  x = tf.reshape(x, [N, C, H, W])
return x * gamma + beta
```

9b illustrates the learned clusters by MFCL when BN is used in the clients' module/decoder. The proposed MFCL(BN) identifies 3 distinguishable clusters for truck, ship, and frog, with some misclassifications within the ship cluster. Specifically, MFCL(BN) fails to accurately separate airplanes from ships. Also, there is a small cluster of automobiles, with the remainder of the automobile samples being very close to the truck cluster.

Figure 9c demonstrates the learned clusters by MFCL when GN is used in the client module. MFCL(GN) can learn 7 distinguishable clusters for truck, ship, frog, automobile, horse, dog, and airplane, with improved separation between the airplane and ship clusters.

Figure 9d shows the learned clusters by MFCL when TN is used in the clients' module/decoder. MFCL(TN) successfully learns 8 distinguishable clusters for truck, ship, frog, automobile, airplane, deer, and horse.

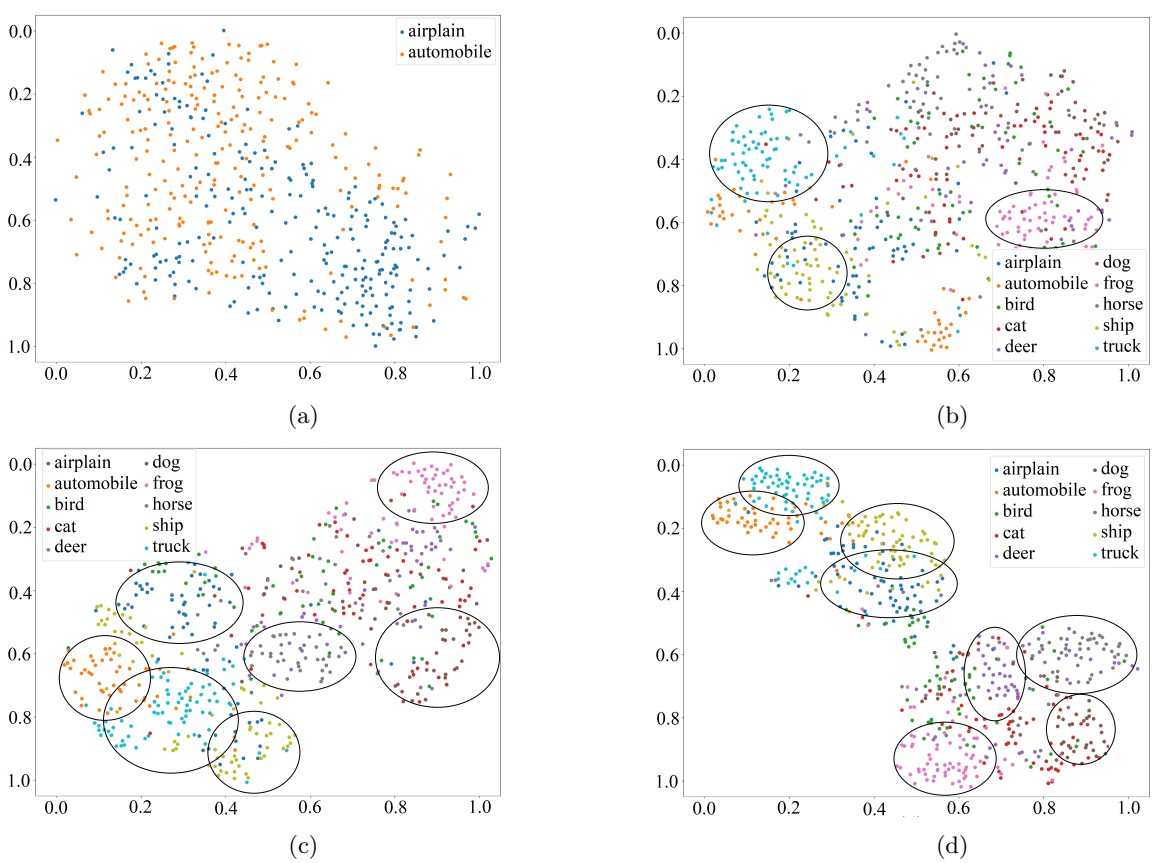

Figure 9: t-SNE visualization of data at a client and the clusters learned from the representation vectors at the rServer in the proposed MFCL. CIFAR-10 is trained with $M = 10$, assuming each client has only 2 classes. (a) t-SNE visualization of raw data in a single client. In the other three figures, t-SNE visualizations of the clusters learned after the projection layer are shown when the normalization technique used in the client module/decoder is (b) BN, (c) GN, and (d) TN.

