# OpenReview forum: "Modular Federated Contrastive Learning with Twin Normalization for Resource-limited Clients"
_TMLR — Accepted by TMLR_

### Review · Reviewer_YJvQ · 2024-08-15

**Summary Of Contributions:**

The paper introduces Modular Federated Contrastive Learning (MFCL) to address resource constraints and data heterogeneity in federated learning. MFCL trains only the initial layers of the model federatedly, reducing communication overhead. It also introduces Peer Normalization (PN) to handle class-imbalanced data more effectively than traditional normalization methods. Experimental results demonstrate that MFCL with PN achieves superior accuracy and efficiency, particularly in heterogeneous data scenarios, compared to conventional methods. The approach also enhances data privacy by separating server functionalities and minimizing data transmission, making it suitable for real-world applications on resource-limited devices.

**Audience:**

No

**Broader Impact Concerns:**

Not involved.

**Claims And Evidence:**

No

**Requested Changes:**

Please see the weakness part.

**Strengths And Weaknesses:**

Strengths:
	The paper is clearly written and well-structured.
	The experimental results are impressive, significantly outperforming the baselines.
	The paper provides a solid background and situates the contribution well within existing research.

Weaknesses:
	Consider adding an algorithm table to outline the steps of the MFCL process. This would enhance the reader’s understanding by providing a clear, step-by-step guide to your methodology.
	Consider adding a paragraph that compares PN with BN and GN. Discussing the similarities and differences will help highlight the incremental improvements and specific advantages of PN in handling heterogeneous data.
	The motivation for training only the initial layers of the model in MFCL isn't clearly explained. Please provide more insight into why this approach was chosen and its advantages over training the entire model.

---

> ### Author Response · Authors · 2024-09-25
>
> Thank you for the comment.
>
> W4.1) We added the algorithm table on page 26 of the revised paper.
>
> W4.2) BN normalizes over the entire mini-batch, which can result in biased and inaccurate statistics when the mini-batch is small or when the data is class-imbalanced. In FL, the challenge is compounded by the fact that clients often have highly class-imbalanced data and average gradients received from the server come from highly non-IID data distributions. This variability in batch statistics across clients makes BN less effective and can lead to degraded performance when the model parameters are aggregated.
> PN, unlike BN, is a sample-based normalization which normalizes two augmented views of one sample and doesn't rely on mini-batches size, which makes it more efficient in class-imbalanced and heterogeneous data settings.
>
> Similar to GN, PN normalizes across a group of channels, making it robust in scenarios with small mini-batches. However, compared to GN, PN leverages more diverse information within each group of channels, as it normalizes the channels of two augmented views of the same sample together. PN effectively captures both inter-example and intra-example relationships, improving model performance.
>
> In summary, while BN struggles with heterogeneous data due to its reliance on batch-wide statistics, and GN, though more robust, lacks inter-example normalization, PN combines the strengths of both. PN retains GN's independence from mini-batch size while having inter-example (not inter-sample) relationships, in a controlled manner by focusing on augmented pairs. This makes PN particularly well-suited for federated SSL in heterogeneous environments, where it improves model performance without suffering from the drawbacks of BN and GN.
>
> Please refer to the second paragraph on page 7.
>
> W4.3) The decision to train only the initial layers of the model in MFCL was made to balance the trade-offs between computational efficiency and communication burden in handling data heterogeneity. This modular approach allows the client module to focus on local representation learning while shifting the resource-intensive parts of training to the server, making it an optimal solution for FL settings with resource-constrained clients and heterogeneous data distributions. We discussed those motivations earlier in the Introduction in the second paragraph on page 3, the bullets related to our contributions on page 3 and in the first paragraph of the section Proposed Modular Federated Contrastive Learning on page 3.
>
> Below, we will provide more insight into why this modular approach was chosen
>
> Resource constraints on clients:
> Client devices in FL are typically resource-constrained, especially in cross-device settings where clients may be smartphones or IoT devices with limited computational power and memory.
> Training the entire model end-to-end on each client would impose a significant computational burden, as deep neural networks, particularly those used in contrastive learning, often require large memory footprints and high computational capacity.
> By training only the initial layers (the client module), we drastically reduce the computational and memory requirements on the client side.
>
> Communication efficiency:
> Traditional FL methods require frequent communication between clients and the server, as the entire model’s parameters are updated and exchanged during each FL round.
> Training only the initial layers of the model on clients allows us to reduce the communication frequency significantly. Instead of sending large model updates (weights or gradients) for the entire model in every FL round, clients only need to send updates for a smaller portion of the model (the initial layers). Each communication event incurs overhead costs, including channel setup,
> data encoding/decoding, handoff management, and session management. High-frequent communication increases latency and network load, potentially leading to delays
> and congestion, especially in bandwidth-limited or high-latency networks like mobile networks. As reported in Table 3, each client utilizing MFCL uses the communication channels 16 times (i.e., 15 FL rounds plus one time to send the representation vectors), which is approximately
> 6 times fewer than FedEMA and 50 times fewer than FedSimCLR.
>
> We discussed those motivations (and benefits) in subsection 5.4 .
>
> Handling heterogeneous and class-imbalanced data:
> By training only the initial layers on the clients, we allow the client module to learn low-level features that are most relevant to the local data of each client. The more task-specific, higher-level features are learned centrally on the server using the representations generated by the client modules. This separation of responsibility allows the model to adapt better to heterogeneous and class-imbalanced data, as the rServer module can focus on learning more generalized, global representations.

---

### Review · Reviewer_nPVs · 2024-08-29

**Summary Of Contributions:**

This paper aims to address the data heterogeneity under resource constraints in federated learning. The authors propose a new method called Modular Federated Contrastive Learning, together with a new normalization method — Peer Normalization. Experiments demonstrate that MFCL outperforms existing methods under severe heterogeneous scenarios.

**Audience:**

Yes

**Broader Impact Concerns:**

N.A.

**Claims And Evidence:**

Yes

**Requested Changes:**

Please improve Figure 2 and addresses some the concerns above.

**Strengths And Weaknesses:**

Strengths:

1. The paper is generally well-structured and easy to follow
2. The problems this paper addresses are important problems in federated learning.
3. The proposed method is interesting and seems to be sound: 1) split the model; 2) use two servers for different operations; 3) peer norm.
4. It is good that the authors explicitly compare with federated split learning.
5. Extensive experiments are provided to evaluate different aspects of the proposed method.

Weakness:

1. The performance of the proposed method seems to be sub-optimal under scenarios that the data heterogeneity is not that severe.
2. Peer norm seems to be the combination of BN and GN. Would it have similar drawbacks of BN in FL?
3. Figure 2 is not easy to understand.
4. Seems that the models evaluated are both ResNet. It would be good to evaluate one alternative backbone.
5. Questions:
    1. What is the goal of this whole learning process? Single global model or personalized models? As the two servers are have no communication, how does it combine and construct a single global model
    2. Why it is named Peer Norm? The name gives a sense that the normalization is between clients (peers) under the context of distributed training.

---

> ### Author Response · Authors · 2024-09-25
>
> Thank you for the comments.
>
> W3.1) In ``real-world scenarios'', it is (much) more practical to assume (severe) heterogeneous and class-imbalanced data scenarios in the clients, which are common challenges in real-world FL applications. MFCL is particularly well-suited for cross-device scenarios with {\it resource-limited clients} who cannot have a large number of samples, which however is typically necessary to represent all (or most) classes. In these cases, therefore, the data distribution tends to be more heterogeneous and class-imbalanced compared to cross-silo scenarios, where clients typically have access to larger datasets with more samples.
>
> W3.2) Please note that PN is not a combination of BN and GN. BN relies on the whole mini-batch, and the size of the mini-batch plays an important role in the success of BN. PN, unlike BN, is a sample-based normalization which normalizes two augmented views or two augmented examples of one sample and doesn't rely on mini-batches size, which makes it more efficient in class-imbalanced and heterogeneous data settings. Specifically, PN is designed to avoid some of the key drawbacks of BN in the context of FL, especially in heterogeneous and class-imbalanced scenarios.  Please note that, in neither BN nor GN, two augmented examples of one sample are normalized together as in PN.
>
> BN in FL and even in centralized learning is sensitive to mini-batch size. This is because BN normalizes over the entire mini-batch, which can result in biased and inaccurate statistics when the mini-batch is small or when the data is class-imbalanced. In FL, the challenge is compounded by the fact that clients not only often have highly class-imbalanced data but also data is severely heterogeneous across the clients, and the average gradients received from the server come from the networks trained on highly heterogeneous data distributions. This variability in batch statistics across clients makes BN less effective and can lead to degraded performance when the model parameters are aggregated.
>
> PN overcomes BN's drawbacks as PN happens over augmented views, not entire mini-batches. Unlike BN, which normalizes over the entire mini-batch, PN focuses on normalizing two augmented views of the same sample together. This ensures that normalization is independent of the mini-batch size and reduces the impact of class-imbalanced and heterogeneous data within the mini-batch.
>
> GN normalizes across a group of channels, making it robust in scenarios with small mini-batches.
> Compared to GN, PN has more diverse information per group of channels as PN normalizes the
> group of channels of two augmented views of the same sample together, making the mean of
> those groups more accurate. PN effectively captures both inter-example and intra-example
> relationships, improving model performance.
>
> We have provided a more detailed discussion of BN, GN, and PN in subsection 3.4 of the revised paper.
>
> W3.3) In the revised paper, we improved Figure 2 and its caption to make it easier to follow.
>
> W3.4) We updated the simulation results for the VGG-16-BN backbone in Table 15 and subsection B.2.3 of the revised paper.
>
> W3.5) In MFCL, the client module is trained to be part of the global model. The rServer module is also trained on the representations extracted from the global client modules. At the end of training, these two components form a single unified global model, which has been our goal for MFCL. To deploy this unified model, the (finally trained) rServer module must be broadcast to the clients.
>
> We would like to expand upon the fact that MFCL is designed to result in a single global model at the end of training, which is why we primarily use normalization to address data heterogeneity and reduce the need for model personalization. However, since MFCL benefits from self-supervised learning, it is always possible to fine-tune the entire model on some labelled data and personalize the global model for each client.
>
> Please refer to the fourth paragraph on page 5 of the revised paper.
>
> W3.6) The term peer in PN refers to the peer relationship between two augmented views of the same sample rather than peer clients. In contrastive learning, these augmented views act as ``peers'' and PN exploits the relationship between these views to normalize them together, bring similar (peer) views closer, and push dissimilar ones apart. To avoid any confusion, in the revised paper, we now use Twin Normalization (TN) which means normalizing two different examples of the same root (sample) together.
>
> RC3.1) In the revised paper, we improved Figure 2 and addressed the concerns.

---

### Review · Reviewer_CJAM · 2024-09-09

**Summary Of Contributions:**

This paper introduces a federated learning algorithm, Modular Federated Contrastive Learning (MFCL), designed to reduce communication, computation, and memory overhead compared to traditional federated learning methods. MFCL adopts a modular approach, splitting the global model into two distinct modules that are trained independently across two severs and multiple clients. The authors conduct comprehensive experiments, benchmarking MFCL against various baseline methods across the CIFAR-10, CIFAR-100, and Tiny-ImageNet datasets under multiple experimental conditions.

**Audience:**

Yes

**Claims And Evidence:**

Yes

**Requested Changes:**

Overall, I believe this work requires significant revisions before it can be considered for publication. Below are some suggestions for improvement:

- Thoroughly address the potential privacy leakage issue: A systematic analysis or visualization demonstrating the likelihood of raw data reconstruction from transmitted representations would strengthen the paper's claims. This would provide clearer insight into the extent of the privacy risks associated with MFCL.

- Ensure claims are well-supported by evidence: Avoid making conclusions without sufficient backing. For example, in the introduction, the paper states, “Self-supervised loss functions… require deep and wide networks. Therefore, extracting representations from the client’s local data using shallow or pruned narrow networks with self-supervised loss functions is impractical.” The reasoning in the first sentence does not adequately support the conclusion in the second. More evidence or clarification is needed to justify this claim.

- Improve the clarity and completeness of the writing: For instance, in the introduction, it would be helpful to include a brief explanation of the proposed Peer Normalization technique, giving readers a preview of the method’s technical details.

- Include error bars in the experimental results: The absence of error bars in the results table is a significant omission. Running multiple experiments and including error bars would help account for randomness and provide more robust and reliable conclusions. This is essential for demonstrating the reproducibility of the findings.

**Strengths And Weaknesses:**

Strengths:

- The paper is clearly written, making it easy to understand and follow.
- The problem addressed is practical and important, focusing on real-world challenges in federated learning.
- The paper includes a dedicated section that effectively compares MFCL with traditional Federated Split Learning, offering valuable insights into their differences.


Weaknesses:

- Privacy Concerns: MFCL involves transmitting data representations, which raises potential privacy issues, as these representations could be used to reconstruct sensitive user data on the server. Although the authors suggest methods to mitigate this risk, the paper lacks a thorough analysis or visual demonstrations to assess how likely such reconstructions are.

- Technical Contribution: The peer normalization method proposed in the paper seems to be a relatively simple modification of batch and group normalization techniques. Given this, it may not be considered a substantial technical contribution in terms of novelty.

---

> ### Author Response · Authors · 2024-09-25
>
> Thank you for the comments.
> W2.1) We have considered two levels of privacy protection in the design of MFCL.
> First, the two-server architecture—where the gServer handles model parameters and the rServer processes only representations—makes a layer of privacy protection by ensuring no single server has access to both representations and model updates.
> Second, in the event of an adversarial attack aimed at data reconstruction, the best-case scenario for the attacker would be the reconstruction of harshly augmented data samples. To further mitigate the risk of reconstruction, MFCL can employ Secure Aggregation (SA) on the gServer and DataMix on the rServer, making data reconstruction significantly more difficult. The simulation results supporting this are presented in Table 6.
> To address the reviewer’s concern, we explored various attack scenarios to assess the potential for data reconstruction from the representations. We provide visual demonstrations of these scenarios in Table 8 on page 14. Please refer to subsection 5.6.2 on page 13 of the revised paper.
>
> W2.2) Please note that PN is not similar to BN or GN. The value of PN lies in its targeted solution to a specific and significant challenge in FL: handling heterogeneous and class-imbalanced data. Unlike BN and GN, which have sub-optimal performance in such scenarios, PN leverages the unique structure of augmented pairs to improve feature representation under severe data heterogeneity. Please note that, in neither BN nor GN, two augmented examples of one sample are normalized together as in PN.
> The effectiveness of PN is demonstrated through substantial performance improvements in the experiments, as shown in our results. As demonstrated in Figure 9 and Table 2 of the paper, PN significantly improves the quality of learned features, ensuring better separation of clusters in highly imbalanced datasets. This clear improvement in cluster separation highlights PN’s effectiveness and justifies its contribution as a practical and impactful method. While PN is conceptually simple, its ability to outperform existing normalization methods in challenging FL scenarios underscores its technical relevance and novelty. Furthermore, its simplicity enables PN to be readily implemented in a variety of scenarios including complexity-limited ones.
>
> RC2.1) To address the reviewer’s concern, we explored various attack scenarios to assess the potential for data reconstruction from the representations. We provide visual demonstrations of these scenarios in Table 8 on page 14. Please refer to subsection 5.6.2 on page 13 of the revised paper.
>
> RC2.2) To avoid any confusion, we have rewritten the sentence as follows: “Shallow or narrow networks are typically insufficient for self-supervised tasks because they lack the capacity to learn rich, informative representations from complex or highly diverse data, particularly in scenarios involving heterogeneous or class-imbalanced data. In FL, where data heterogeneity is a major challenge, shallow networks are unlikely to extract representations that generalize well across different clients, especially when each client has unique data distributions. Self-supervised learning methods require networks with enough depth and capacity to align representations from diverse data sources effectively.” Please refer to the first paragraph on page 2.
>
> Please note that the above discussion is based on established research results showing that self-supervised learning, particularly contrastive learning, benefits from wider models and large datasets. For example, it is supported by Figure 1 of SimCLR paper in which “SimCLR (4x)” refers to an enlarged version of the standard ResNet-50 architecture used in the experiments. Specifically, “4x” means that the width of the network has been increased by a factor of 4, which significantly increases the capacity of the model. Self-supervised methods, such as contrastive learning, rely on learning representations by maximizing agreement between different augmented views of the same data. To achieve high performance, such methods require networks with high capacity (i.e., deep and wide networks) to capture the necessary complexity and nuances in the data. Deep and wide networks help capture more abstract features and better generalize over complex data distributions, especially when applied in non-IID settings, which is common in FL scenarios.
>
> RC2.3) In the revised paper, we include a concise summary of PN, highlighting its key idea of normalizing augmented pairs together to enhance feature consistency under heterogeneous and class-imbalanced data conditions. Please refer to the second paragraph on page 3 of the revised paper.
>
> RC2.4) As was also reported in the paper, ``each scenario is run 3 times and the mean of the top-1 accuracy on a uniform test set is reported''. In the revised paper, we included the errors in Table 2.

---

### Review · Reviewer_5GpL · 2024-09-10

**Summary Of Contributions:**

The paper proposes a modular FL approach to address the compute/communication cost of self-supervised contrastive FL frameworks, mostly imposed on the clients involved. Here model parameters are divides into two chunks and trained independently through two separate servers. The paper also proposes a form of normalization tailored towards their approach to addressing data heterogeneity in a contrastive setting. The outcome is compared against a suite of self-supervised and fully supervised FL methods.

**Audience:**

Yes

**Claims And Evidence:**

Yes

**Requested Changes:**

All requested changes are already listed in the weaknesses section.

**Strengths And Weaknesses:**

**Strengths**
- The paper is well written and the narrative is coherent.
- They tackle two important problems in Federated SSL at once, (i) compute/comms cost and (ii) data heterogeneity.
- Differences against some of the well-known competitor baseline are properly sketched and elaborated which helps understanding the core advantages of the proposed approach.

**Weaknesses**
- Comparison against a few other relevant and *most recent* baselines are missing (most references are up until 2022). Here are a few other ones to consider:
    - [for Federated SSL] Does Learning from Decentralized Non-IID Unlabeled Data Benefit from Self Supervision? (ICLR 2023)
    - [for Federated SSL] A Mutual Information Perspective on Federated Contrastive Learning (ICLR 2024)
- On a related note, there are other FL baselines that are communication/compute efficient. They do not communicate all the model parameters or representations of the entire local dataset. Only prototypes of the local data are used to facilitate collaboration. Examples include:
    - Personalized Federated Learning with Feature Alignment and Classifier Collaboration (ICLR 2023)
    - Conditional Moment Alignment for Improved Generalization in Federated Learning (NeurIPS FL workshop 2022)

- Peer normalization (PN) seems to be a combination of Batch and Group normalizations. This has to be further elaborated in the paper. What are the pros and cons of adopting such a construct? Is this only beneficial in Federated SSL, and why is it called "peer" in the sense that it hardly related to the peer clients, and more so to the SSL training?

- It is unclear how the two model parts/components come together at the end of training (single global or multiple personalized ones)? The fact that these two components are trained separately might lead to communication efficiency, but what is the impact on the performance? Is there some sort of trade off here? These aspects need to be further clarified in the paper.

- Minor one, on page 4, $\theta_m^t$ denotes the encoder and decoder parameter. However, the input and output shape of each component is different. Therefore, given the parameters of the encoder is $\theta_m^t$, the decoder parameters should be the transpose $[\theta_m^t]^T$. Nonetheless, it is not clear if the parameters are shared or if they are replaced at each round to make them equal?

---

> ### Author Response · Authors · 2024-09-25
>
> Thank you for the comments.
> W1.1) Both papers are indeed interesting. The first approach focuses on decentralized self-supervised learning and provides insights into how decentralized SSL is robust to the heterogeneity of decentralized datasets, enabling the learning of useful representations for object classification. This aligns well with our claim that SSL is more robust to data heterogeneity in FL.
> The second paper aims at maximizing mutual information between different views of data, while also dealing with different types of non-IIDness.
> While the second paper presents promising results, it is not directly comparable to MFCL, as it leverages labelled data. Additionally, we were unable to locate a publicly available implementation of the code. We are currently working on reproducing the code based on the details provided in the paper and will update Table 2 accordingly.
> Please see Table 2 and the last paragraph on page 8.
>
> W1.2) We did not consider these papers earlier because their main focus is on personalization. However, in response to the reviewer's concerns, we have now included the results from FedPAC papers in our work. Please see Table 4 and the fourth paragraph on page 11.
>
> W1.3)
>  -PN is not a simple combination of BN and GN. With some inspiration from BN and GN, PN is uniquely adapted for contrastive learning. Specifically,
> 1) BN relies on the whole mini-batch. PN, unlike BN, is a sample-based normalization which normalizes two augmented views of one sample and doesn't rely on mini-batches size, which makes it more efficient in class-imbalanced and heterogeneous data settings.
>
> 2) GN normalizes across a group of channels, making it robust in scenarios with small mini-batches. Compared to GN, PN has more diverse information per group of channels as PN normalizes the group of channels of two augmented views of the same sample together. PN effectively captures both inter-example and intra-example relationships, improving model performance.
> 3) In neither BN nor GN, two augmented examples of one sample are normalized together as in PN.
> We have provided a more detailed discussion of BN, GN, and PN in subsection 3.4 on page 6.
>
> Pros:
> 1) Improved performance on heterogeneous and class-imbalanced data
> 2) Low dependency on mini-batch size
>
> Cons:
> 1) Less established than BN and GN
> 2) Less effective in class-balanced or homogeneous data
> Please refer to the third and fourth paragraphs on page 6 of the revised paper.
>
> -While PN is particularly beneficial in ``Federated SSL'', it is not necessarily limited to this context. PN could be extended to any learning tasks that benefit from augmented data samples and the data is class-imbalanced.
> Please refer to the footnote on page 7.
>
> -peer in PN refers to the peer relationship between two augmented views of the same sample rather than peer clients. In contrastive learning, these augmented views act as ``peers'' and PN exploits the relationship between these views to normalize them together, bring similar (peer) views closer, and push dissimilar ones apart. To avoid any confusion, in the revised paper, we now use the term Twin Normalization (TN).
>
> W1.4)
> -In MFCL, the client module is trained to be part of the global model. The rServer module is also trained on the representations extracted from the global client modules. At the end of training, these two components form a single unified global model, which has been our goal for MFCL. To deploy this unified model, the (finally trained) rServer module must be broadcast to the clients.
> Please refer to the fourth paragraph on page 5.
>
> -Second, based on the simulation results in Table 2, MFCL improves the performance of FL in severe heterogeneous and class-imbalanced data distributions and excels the other end-to-end self-supervised learning methods. However, in scenarios with homogeneous and class-balanced data, end-to-end supervised learning learning still performs the best.
>
> We provided a detailed discussion of accuracy performance and trade-off of MFCL in subsection 5.3 on page 9. We also discussed the communication burden of MFCL in subsection 5.4 on page 10.
>
> Finally, we would like to expand upon the fact that MFCL is designed to result in a single global model at the end of training, which is why we primarily use normalization to address data heterogeneity and reduce the need for model personalization. However, since MFCL benefits from self-supervised learning, it is always possible to fine-tune the entire model on some labelled data and personalize the global model for each client.
>
> W1.5) The encoder and decoder indeed have different input and output shapes, as the encoder reduces the dimensionality of the input and the decoder reconstructs it. In the revised paper, we corrected the notation.

---

### Decision · Action_Editor_6YRf · 2024-10-28

**Recommendation:** Accept as is

**Comment:**

This paper studies a modular framework for federated self-supervised learning. The proposal, called Modular Federated Contrastive Learning (MFCL), trains the first few layers through a server in a federated fashion, and then trains other layers at a local server with no communication. The authors additionally propose Peer Normalization (PN) to tackle data heterogeneity in this context. There were several questions/comments about privacy leakage, the presentation of the results, as well as effectiveness of peer normalization by the reviewers that the authors have successfully addressed through additional evidence. As such, the paper is recommended to be accepted. Congratulations to the authors!

**Audience:**

The paper is of interest to a broad audience in machine learning.

**Claims And Evidence:**

The claims are substantiated by numerical evidence.